# TAGGING THE THOUGHT: UNLOCKING PERSONALIZATION REASONING VIA REINFORCEMENT LEARNING

## ABSTRACT

Recent advancements have endowed Large Language Models (LLMs) with impressive general reasoning capabilities, yet they often struggle with personalization reasoning—the crucial ability to analyze user history, infer unique preferences, and generate tailored responses. To address this limitation, we introduce **TagPR**, a novel training framework that significantly enhances an LLM's intrinsic capacity for personalization reasoning through a "tagging the thought" approach. Our method first develops a data-driven pipeline to automatically generate and semantically label reasoning chains, creating a structured dataset that fosters interpretable reasoning. We then propose a synergistic training strategy that begins with Supervised Fine-Tuning (SFT) on this tagged data to establish foundational reasoning patterns, followed by a multi-stage reinforcement learning (RL) process. This RL phase is guided by a unique composite reward signal, which integrates tag-based constraints and a novel Personalization Reward Model with User Embeddings (PRMU) to achieve fine-grained alignment with user-specific logic. Extensive experiments on the public LaMP benchmark and a self-constructed dataset demonstrate that our approach achieves state-of-the-art results, delivering an average improvement of 32.65% over the base model across all tasks. Our work validates that structured, interpretable reasoning is a highly effective pathway to unlocking genuine personalization capabilities in LLMs.[1]

## 1 INTRODUCTION

While Large Language Models (LLMs) have demonstrated remarkable proficiency in general reasoning tasks such as mathematics and coding (Guo et al., 2025; Yu et al., 2025), their success does not readily translate to personalization—a domain crucial for creating truly user-centric applications, from recommendation engines to bespoke conversational agents. Effective personalization demands more than generic logic; it requires personalization reasoning: the ability to meticulously analyze a user's historical data, infer their unique preferences and idiosyncratic thought patterns, and synthesize this understanding to generate a tailored response.

Surprisingly, even the most powerful reasoning-centric LLMs falter in this area, often failing to outperform standard models on personalization benchmarks. This performance gap arises from a fundamental misalignment: models optimized for general-purpose reasoning tend to prioritize their own internal, generalized logic over the specific, often divergent, context provided by a user's profile. This leads to responses that are generic or, worse, contradictory to the user's established preferences. Pioneering studies such as R2P (Luo et al., 2025b) and RPM (Kim et al., 2025) have highlighted this very issue. While these methods have made progress by guiding models with templates or preconstructed reasoning paths, they often act as external scaffolds rather than fundamentally enhancing the model's intrinsic ability to reason about a user.

Our core motivation stems from the observation that personalization reasoning is not a monolithic act of intuition, but a structured, multi-step process of analyzing user history, identifying recurring patterns, and applying those patterns to new contexts. The opaque, free-form reasoning of standard LLMs is ill-suited to this procedural task. Furthermore, existing RL-based approaches for eliciting personalization reasoning are often hampered by their reliance on outcome-based rewards. Such signals are inherently sparse and indirect, as a correct final answer does not necessarily imply a valid

---

[1]All code is included in the Supplementary Material.

Figure 1: A comparison of reasoning paths. Left: The Generalization Model (Qwen3-8B) uses free-form logic, leading to an incorrect tag ("social commentary"). Right: Our Personalization Model follows a structured path to correctly infer the user-specific tag ("dystopia").

reasoning process. We argue that forcing a model to follow an explicit, structured workflow is key to unlocking its personalization potential. To this end, we introduce **TagPR**, a novel framework centered on "tagging the thought". By transforming implicit reasoning steps into an explicit structure, our approach enables precise process-level supervision. Instead of allowing the model to reason implicitly, we compel it to externalize its logic into a sequence of discrete, interpretable steps, each marked with a semantic tag (e.g., <examine_examples>, <identify_patterns>). These tags act as cognitive waypoints, transforming the complex task of personalization into a manageable, explicit procedure that the model can learn to execute robustly, as illustrated in Figure 1.

This is achieved through a synergistic training strategy. First, we pioneer a data-driven pipeline to automatically generate a new dataset of reasoning chains labeled with these semantic tags. We use this dataset for Supervised Fine-Tuning (SFT) to instill the foundational grammar of structured, personalized thought. Following this, we employ a multi-stage reinforcement learning (RL) process to refine this capability. This RL phase is guided by a novel composite reward that combines tag-based structural constraints with a fine-grained signal from our new Personalization Reward Model with User Embeddings (PRMU), which explicitly aligns the model's reasoning with user-specific logic. Our key contributions are threefold:

I. We pioneer a data-driven pipeline to automatically generate and label reasoning chains with semantic tags, creating a new dataset to foster structured, interpretable reasoning. This dataset will be made publicly available to facilitate future research.

II. We introduce a synergistic SFT and multi-stage RL training framework. This process is guided by a unique composite reward signal that integrates tag-based constraints and our novel Personalization Reward Model with User Embeddings (PRMU) for fine-grained alignment with user logic.

III. We demonstrate through extensive experiments on the public LaMP, LongLaMP, PGraphRAG benchmark and a self-constructed dataset that our approach, **TagPR**, achieves state-of-the-art results, significantly outperforming strong baselines and even larger proprietary models, thereby effectively unlocking superior personalization reasoning.

## 2 RELATED WORK

**Reasoning Enhancement through Reinforcement Learning** Recent advances in large language models have significantly improved reasoning capabilities through sophisticated reinforcement learning techniques. Building on foundational algorithms like PPO (Schulman et al., 2017), newer methods such as Group Relative Policy Optimization (GRPO) (Shao et al., 2024) have been instrumental in training advanced reasoning models like DeepSeek-R1 (Guo et al., 2025). This line of work has been extended by innovations including DAPO (Yu et al., 2025) for improving long chains of thought generation, and Group Sequence Policy Optimization (GSPO) (Zheng et al., 2025a) for sequence-level optimization with enhanced stability. These RL methods have proven particularly effective in specialized domains: Search-R1 (Jin et al., 2025) and DeepResearcher (Zheng et al., 2025b) enhances reasoning for web-based question answering, ReTool (Feng et al., 2025) targets mathematical reasoning through code execution, GUI-R1 (Luo et al., 2025a) develops reasoning

for graphical task automation, and DeepEyes (Zheng et al., 2025c) integrates visual reasoning, and Intern-S1 (Bai et al., 2025) focuses on scientific reasoning domains.

**Large Language Model Personalization** LLM personalization has evolved rapidly since the establishment of foundational benchmarks like LaMP (Salemi et al., 2024b). A dominant approach is retrieval-augmented generation, with innovations including feedback-optimized retrieval (Salemi et al., 2024a) and generation-calibrated retrievers (Mysore et al., 2024). PAG (Richardson et al., 2023) enhances retrieval by integrating user history summarization. Beyond retrieval, research has explored core personalization components (Wu et al., 2024), revealing that historical responses are more critical than semantic content alone DPL (Qiu et al., 2025) models inter-user differences to capture unique preferences. Parameter-efficient approaches include OPPU (Tan et al., 2024b) with user-specific lightweight modules, PER-PCS (Tan et al., 2024a) for collaborative PEFT sharing, direct parameter injection (Zhang et al., 2024), plug-and-play user embeddings (PPlug) (Liu et al., 2024), and HYDRA (Zhuang et al., 2024) for black-box personalization. Additional methods include multi-stage decomposition (Li et al., 2023) and multi-objective parameter merging (P-Soups) (Jang et al., 2023).

**Personalization Reasoning** Personalization reasoning represents an emerging intersection of reasoning capabilities and personalization tasks. Early approaches primarily use prompting strategies for black-box models: RPM (Kim et al., 2025) constructs individualized reasoning paths from user history, while R2P (Luo et al., 2025b) employs hierarchical reasoning templates. Fine-tuning approaches include generating reasoning paths followed by iterative self-training (Salemi et al., 2025), and reinforcement learning for preference inference through extended inductive reasoning (Li et al., 2025). Most closely related to our work, PrLM (Zhang et al., 2025) uses contrastive reward models with reinforcement learning for reasoning in personalization generation tasks. While these methods have made notable progress, they typically address personalization reasoning through either template-guided generation or reward-based optimization without fundamentally restructuring how models approach the multi-faceted nature of personalization tasks. Our work introduces a novel paradigm that combines structured semantic tagging with specialized reward modeling to unlock the model's intrinsic capacity for structured personalization reasoning.

# 3 METHODOLOGY

This section presents the methodology for **TagPR**. We begin by formulating the task in Section 3.1 and detailing our data construction pipeline in Section 3.2. Subsequently, we introduce the Personalization Reward Model (PRMU) in Section 3.3 and our three-stage training strategy, which progresses from SFT to a two-stage RL refinement in Section 3.4.

## 3.1 TASK FORMULATION

We define personalized reasoning as the task of generating a user-specific response $y$ to a query $x$, conditioned on the user's profile $P_u = \{(x_i, y_i)\}_{i=1}^{N_u}$, which consists of their historical interactions.

Our approach enhances this process by first generating an explicit reasoning chain $c$ before producing the final response $y$. Conditioned on the query $x$ and a relevant profile subset $p_u \subseteq P_u$, our model (parameterized by $\theta$) is trained to maximize the joint likelihood of the chain and response:

$$p(c, y|x, p_u; \theta) = p(c|x, p_u; \theta) \cdot p(y|c, x, p_u; \theta). \tag{1}$$

The core challenge is to ensure the reasoning chain $c$ is coherent and faithful to the user's profile $p_u$, and that the response $y$ remains consistent with this explicit reasoning.

## 3.2 TAGGED REASONING CHAINS CONSTRUCTION PIPELINE

To facilitate the generation of explicitly tagged reasoning steps in large language models, we designed a multi-stage pipeline to construct a high-quality dataset for SFT. This pipeline, illustrated in Figure 2, systematically generates, filters, and annotates reasoning chains, culminating in a final dataset of approximately 10,000 instances. The process is organized into three primary stages:

**Raw Reasoning Chain Generation.** The pipeline commences with data sampling from the LaMP dataset (Salemi et al., 2024b), a benchmark for personalization tasks. We randomly selected 1,000

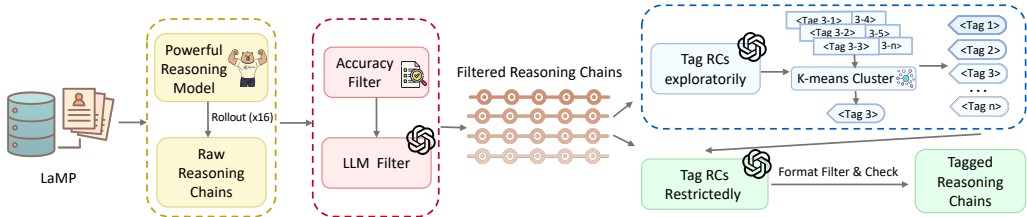

Figure 2: The pipeline for constructing our Tagged Reasoning Chains dataset. The process includes raw chains generation from LaMP, a two-stage quality filter, and a two-phase tagging procedure where primary tags are first defined via clustering and then applied in a restricted final annotation.

instances from each of its six training tasks. For each instance, we employed a powerful reasoning model, Qwen3-235B-A22B-Thinking-2507 (Team, 2025), to generate 16 candidate reasoning chains via rollout, thereby creating a diverse initial pool of raw reasoning chains.

**Two-Stage Filtering.** To ensure the integrity and quality of the dataset, we implemented a rigorous two-stage filtering protocol. First, an *accuracy filter* was applied to retain only correctly answered samples. For classification tasks (LaMP-1, LaMP-2, LaMP-3), this involved verifying the final prediction against the ground truth. For generation tasks (LaMP-4, LaMP-5, LaMP-7), we calculated the ROUGE score (Lin, 2004) and preserved only samples that surpassed a predetermined threshold. Second, the accuracy-filtered chains were subjected to an *LLM filter*, where GPT-4o (Hurst et al., 2024) scored each chain based on qualitative metrics such as logical consistency, factual accuracy, completeness, and conciseness. Only instances achieving a composite score greater than 15 were retained for the tagging stage (the detailed prompt is provided in the Appendix F.1.1).

**Two-Phase Tagging** The filtered reasoning chains (RCs) then underwent a two-phase tagging procedure to assign meaningful and consistent tags. In the first phase, *exploratory tagging*, we prompted GPT-4o to perform unrestricted tagging on the RCs, generating a wide range of descriptive tags. These preliminary tags were then semantically clustered using the K-means algorithm (MacQueen, 1967). This unsupervised method allowed us to group similar tags and identify high-frequency, salient reasoning patterns, resulting in a refined set of 9 primary tags: `<analyze_input>`, `<examine_examples>`, `<identify_patterns>`, `<evaluate_reference>`, `<compare_entities>`, `<synthesize_findings>`, `<make_decision>`, `<verify_conclusion>` and `<formulate_conclusion>`. In the second phase, *restricted tagging*, the reasoning chains were re-annotated by GPT-4o, but this time constrained to use only the 9 established primary tags. This step ensured consistency and correctness across the entire dataset. Finally, the re-tagged data underwent an automated format filter and a manual sampling check to guarantee quality. This meticulous pipeline yielded our final dataset of approximately 10,000 high-quality, tagged reasoning chains ready for model fine-tuning. Detailed tagging prompts are provided in the Appendix F.1.2 and Appendix F.1.3.

### 3.3 PERSONALIZATION REWARD MODEL WITH USER EMBEDDINGS

To overcome the limitations of generic reward models, we introduce the **Personalization Reward Model with User Embeddings (PRMU)**. Unlike standard architectures, PRMU incorporates learnable user embeddings $E_u$ to capture individual preferences. This architectural modification enables it to provide a granular reward signal that prioritizes reasoning which is not only accurate but also highly tailored to the user's profile, guiding the model towards genuinely personalized responses.

PRMU is trained on two bespoke preference datasets ($\sim$10k samples each). The **Profile-Reasoning Preference (PRP)** dataset contrasts responses generated with a user profile (preferred) against those generated without (rejected), teaching the model to value profile utilization. The **Personalized-Quality Preference (PQP)** dataset contains pairs of personalized responses where preference is determined by correctness or ROUGE score, thereby training the model to discern reasoning quality.

Initialized from Skywork-Reward-V2-Qwen3-0.6B (Liu et al., 2025), our PRMU architecture first maps a user ID $id_u$ to its corresponding embedding $E_u$. This embedding, along with the query, profile, and reasoning chain, is processed to produce a scalar logit. Both the base reward model param-

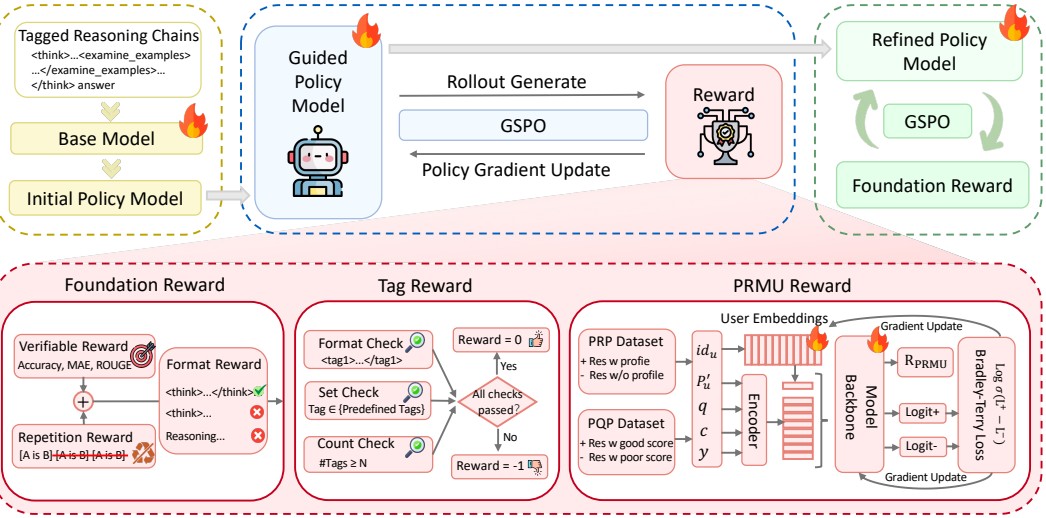

Figure 3: Overview of our proposed multi-stage training framework. An initial policy model is obtained via SFT on tagged reasoning chains. The model is then refined through two sequential RL phases: (1) a **Guided RL** stage using a complex, multi-component reward (including Tag and PRMU rewards) to learn structured reasoning, and (2) an **Exploratory RL** stage with a Foundation reward to further boost performance.

eters $\theta_r$ and the user embeddings $E$ are jointly optimized by minimizing the Bradley-Terry (Bradley & Terry, 1952) preference loss:

$$\mathcal{J}(\theta_r, E) = -\mathbb{E}_{(x^+, x^-) \sim \mathcal{D}} \left[ \log \sigma \left( f_{\text{PRMU}}(x^+) - f_{\text{PRMU}}(x^-) \right) \right] \tag{2}$$

where $x^+$ and $x^-$ represent the preferred and rejected input tuples from our preference dataset $\mathcal{D}$. The model's final output is transformed by a sigmoid function to yield the normalized reward score, $R_{\text{PRMU}}$, for the reinforcement learning phase:

$$R_{\text{PRMU}} = \sigma(f_{\text{PRMU}}(id_u, q, p_u, c, y | E_u; \theta_r)). \tag{3}$$

## 3.4 FROM SFT TO TWO-STAGE RL

As illustrated in Figure 3, our training pipeline progresses from SFT through a two-stage RL process designed to first instill structured reasoning and then refine performance.

**Foundational SFT for Knowledge Bootstrapping** We begin by fine-tuning a base model on our labeled reasoning chains dataset. This SFT stage bootstraps the model with the fundamental knowledge of reasoning with tags. The objective is to maximize the conditional log-likelihood of generating the reasoning chain $c$ and answer $y$ given a query $q$ and user profile $p_u$:

$$\mathcal{J}_{\text{SFT}}(\theta) = - \sum_{(q, p_u, c, y) \in \mathcal{D}} \log P_\theta(c, y | q, p_u), \tag{4}$$

where $\mathcal{D}$ is the labeled dataset and $\theta$ are the model parameters. This produces an initial policy model capable of tagged reasoning, albeit at a preliminary level.

**Guided RL for Personalization Reasoning** Following SFT, we initiate a guided RL stage to enhance the model's personalized reasoning capabilities. We design a comprehensive reward function, $R$, as a weighted combination of five distinct signals:

$$R = \alpha \cdot (R_v + R_{\text{rep}}) \cdot R_f + \beta \cdot R_{\text{tag}} + \gamma \cdot R_{\text{PRMU}}, \tag{5}$$

where we set the balancing hyperparameters $\alpha = \beta = 0.8$ and $\gamma = 0.2$. These values prioritize logical correctness over personalization to prevent factuality degradation and reward hacking. We specifically adopt a multiplicative form for $R_f$ to condition other signals on format compliance. We include a sensitivity analysis in the Appendix C, which confirms that our method remains robust

across a reasonable range of hyperparameter variations, with the current configuration yielding optimal balance. The Personalization Reward $R_{\text{PRMU}}$ is introduced in Section 3.3. Other components are defined as follows.

Verifiable Reward ($R_v$) measures the factual correctness of the response $y$ against a ground-truth reference $y^*$:

$$R_v(y, y^*) = \begin{cases} \text{Accuracy}(y, y^*) & \text{for classification tasks} \\ \text{ROUGE}(y, y^*) & \text{for generation tasks} \end{cases}. \tag{6}$$

Format Reward ($R_f$) provides a binary signal to enforce structural integrity:

$$R_f(c, y) = \begin{cases} 1 & \text{if } c, y \text{ match the expected format} \\ 0 & \text{otherwise} \end{cases}. \tag{7}$$

Repetition Reward ($R_{\text{rep}}$) penalizes textual redundancy to improve fluency:

$$R_{\text{rep}}(c, y) = -\frac{|T_n(c, y)| - |U_n(c, y)|}{|T_n(c, y)| + \delta}, \tag{8}$$

where $T_n$ and $U_n$ are the multiset and set of n-grams in the generation respectively, and $\delta$ is a small constant for stability.

Tag Reward ($R_{\text{tag}}$) enforces the structural and semantic correctness of the tagged reasoning. It is a penalty-based signal:

$$R_{\text{tag}}(c, y) = \begin{cases} 0 & \text{if all logical checks on } c, y \text{ pass} \\ -1 & \text{otherwise} \end{cases}. \tag{9}$$

The checks include verifying tag format, ensuring tags belong to a predefined set, and meeting a minimum tag count.

For policy optimization, we utilize the GSPO algorithm, which offers greater training stability by operating at the sequence level. The GSPO objective is:

$$\mathcal{J}_{\text{GSPO}}(\theta) = \mathbb{E}_{\substack{q \sim \mathcal{D} \\ \{c_i, y_i\}_{i=1}^G \sim \pi_{\theta_{\text{old}}}(\cdot|q)}} \left[ \frac{1}{G} \sum_{i=1}^G \min\left( s_i(\theta)\hat{A}_i, \text{clip}(s_i(\theta), 1-\epsilon, 1+\epsilon)\hat{A}_i \right) \right] \tag{10}$$

where $s_i(\theta)$ is the sequence-level importance sampling ratio and $\hat{A}_i$ is the standardized advantage for each response in a generated group of size $G$.

**Exploratory RL for Performance Refinement** In the final stage, we address performance plateaus by introducing an exploratory RL phase. This stage employs a simplified Foundation Reward signal, focusing exclusively on fundamental quality metrics:

$$R_{\text{foundation}} = (R_v + R_{\text{rep}}) \cdot R_f. \tag{11}$$

By removing the personalization and tag reward constraints, this stage encourages the model to freely explore the policy space, further refining its personalized reasoning ability by maximizing core performance.

## 4 EXPERIMENT

### 4.1 EXPERIMENTAL SETUP

**Implementation Details** We employ Qwen3-8B as our base model. Our training process consists of SFT on the dataset described in Section 3.2, followed by a two-stage RL phase using data sampled from the LaMP training set. We evaluate our model on the LaMP benchmark, a standard for assessing personalization, reporting results on its validation set as the test set is not public.

**Baselines** We conduct a comprehensive comparison against a wide spectrum of baselines. These include: (1) standard methodologies such as Zero-shot, RAG, PAG (Richardson et al., 2023), SFT, SFT-Ind, and their reasoning-enhanced variants (-R); (2) advanced personalization (PPlug (Liu et al.,

2024), HYDRA-Adapter (Zhuang et al., 2024)) and reasoning-focused techniques (R2P (Luo et al., 2025b), PrLM (Zhang et al., 2025)); and (3) state-of-the-art large language models like GPT-4o and Gemini-2.5-Pro (Comanici et al., 2025). One primary baseline is the RAG-R method, which shares our configuration with the original Qwen3-8B model. For clarity, we refer to it as **Base** in subsequent sections.

More detailed descriptions of all baselines, hyperparameters, benchmark, evaluation metrics, and experimental configurations are provided in the Appendix A.

Table 1: Main results on the LaMP benchmark, comparing TagPR against a wide range of baselines. **Bold** indicates the best performance, and underline indicates the second-best. The "R" column denotes whether a reasoning step is used (✓).

| Dataset → | | LaMP-1 | | LaMP-2 | | LaMP-3 | | LaMP-4 | | LaMP-5 | | LaMP-7 | |
|---|---|---|---|---|---|---|---|---|---|---|---|---|---|
| Method | R | ACC ↑ | F1 ↑ | ACC ↑ | F1 ↑ | MAE ↓ | RMSE ↓ | R-1 ↑ | R-L ↑ | R-1 ↑ | R-L ↑ | R-1 ↑ | R-L ↑ |
| *Previous Method* | | | | | | | | | | | | | |
| Zero-shot | ✗ | 0.498 | 0.470 | 0.318 | 0.244 | 0.639 | 0.983 | 0.144 | 0.125 | 0.417 | 0.351 | 0.465 | 0.413 |
| Zero-shot-R | ✓ | 0.477 | 0.483 | 0.389 | 0.347 | 0.416 | 0.778 | 0.131 | 0.115 | 0.354 | 0.306 | 0.431 | 0.383 |
| RAG | ✗ | 0.668 | 0.645 | 0.414 | 0.361 | 0.354 | 0.710 | 0.158 | 0.139 | 0.453 | 0.384 | 0.473 | 0.419 |
| RAG-R (Base) | ✓ | 0.717 | 0.722 | 0.453 | 0.413 | 0.291 | 0.645 | 0.152 | 0.137 | 0.434 | 0.365 | 0.439 | 0.391 |
| PAG | ✗ | 0.677 | 0.649 | 0.420 | 0.367 | 0.337 | 0.675 | 0.167 | 0.148 | 0.452 | 0.385 | 0.479 | 0.426 |
| PAG-R | ✓ | 0.731 | 0.736 | 0.470 | 0.417 | 0.289 | 0.627 | 0.160 | 0.142 | 0.408 | 0.349 | 0.428 | 0.380 |
| SFT | ✗ | 0.670 | 0.654 | 0.511 | 0.461 | 0.273 | 0.569 | 0.196 | 0.178 | 0.455 | 0.393 | 0.498 | 0.445 |
| SFT-R | ✓ | 0.722 | 0.724 | 0.456 | 0.416 | 0.339 | 0.878 | 0.159 | 0.145 | 0.440 | 0.378 | 0.437 | 0.386 |
| SFT-Ind | ✗ | 0.717 | 0.717 | 0.532 | 0.488 | 0.269 | 0.568 | 0.207 | 0.187 | 0.463 | 0.411 | 0.507 | 0.454 |
| SFT-Ind-R | ✓ | 0.729 | 0.731 | 0.463 | 0.419 | 0.366 | 1.001 | 0.151 | 0.138 | 0.432 | 0.374 | 0.433 | 0.383 |
| PPlug | ✗ | 0.698 | 0.699 | 0.535 | 0.489 | 0.261 | 0.532 | 0.213 | 0.195 | 0.486 | 0.434 | 0.521 | 0.465 |
| HYDRA-Adapter | ✗ | 0.692 | 0.692 | 0.482 | 0.455 | 0.320 | 0.663 | 0.159 | 0.138 | 0.457 | 0.395 | 0.483 | 0.423 |
| R2P | ✓ | 0.729 | 0.730 | 0.487 | 0.459 | 0.267 | 0.557 | 0.176 | 0.155 | 0.459 | 0.396 | 0.489 | 0.426 |
| PrLM | ✓ | 0.731 | 0.731 | 0.534 | 0.504 | 0.288 | 0.635 | 0.183 | 0.169 | 0.499 | 0.438 | 0.513 | 0.459 |
| *State-of-the-Art LLMs* | | | | | | | | | | | | | |
| GPT-4o | ✗ | 0.733 | 0.733 | 0.542 | 0.512 | 0.254 | 0.554 | 0.191 | 0.175 | 0.470 | 0.407 | 0.475 | 0.419 |
| Qwen3-235B-A22B | ✓ | 0.715 | 0.720 | 0.511 | 0.488 | 0.280 | 0.633 | 0.177 | 0.158 | 0.450 | 0.396 | 0.455 | 0.409 |
| Deepseek-R1 | ✓ | 0.740 | 0.744 | 0.522 | 0.488 | 0.280 | 0.644 | 0.181 | 0.166 | 0.451 | 0.399 | 0.447 | 0.397 |
| Gemini-2.5-Pro | ✓ | 0.761 | 0.761 | 0.582 | 0.548 | 0.271 | 0.594 | 0.222 | 0.202 | 0.495 | 0.438 | 0.480 | 0.425 |
| *Our Method* | | | | | | | | | | | | | |
| TagPR w/o RL | ✓ | 0.722 | 0.724 | 0.456 | 0.416 | 0.339 | 0.878 | 0.159 | 0.145 | 0.440 | 0.378 | 0.437 | 0.386 |
| TagPR w/o SFT | ✓ | 0.747 | 0.747 | 0.543 | 0.510 | 0.271 | 0.593 | 0.194 | 0.181 | 0.502 | 0.441 | 0.525 | 0.469 |
| TagPR w/o Tag | ✓ | 0.749 | 0.749 | 0.545 | 0.511 | 0.272 | 0.595 | 0.197 | 0.183 | 0.506 | 0.441 | 0.524 | 0.469 |
| TagPR w/o Reward | ✓ | 0.768 | 0.769 | 0.557 | 0.514 | 0.246 | 0.393 | 0.205 | 0.190 | 0.522 | 0.453 | 0.545 | 0.490 |
| **TagPR** | ✓ | **0.803** | **0.803** | **0.598** | **0.557** | **0.218** | **0.263** | **0.234** | **0.213** | **0.542** | **0.471** | **0.565** | **0.507** |

## 4.2 MAIN RESULTS

The results, presented in Table 1, demonstrate that **TagPR** establishes a new state-of-the-art across all six tasks of the LaMP benchmark. It consistently outperforms a comprehensive suite of baselines, including prior personalization methods, reasoning-focused models, and even substantially larger proprietary LLMs.

To isolate the efficacy of our framework, we first conduct an ablation study comparing **TagPR** against a **Base** (RAG-R) method. This baseline shares an identical configuration but utilizes the original Qwen3-8B model. The performance gains are substantial: **TagPR** achieves a 55.5% relative improvement in ROUGE-L on the LaMP-4 generation task, boosts the F1-score by 34.9% on the challenging LaMP-2 classification task, and reduces the MAE by 25.1% on the LaMP-3 task. These results underscore that our synergistic training paradigm significantly enhances the model's personalization reasoning capabilities.

Notably, our fine-tuned 8B parameter model consistently outperforms leading proprietary models that are orders of magnitude larger. For instance, on the LaMP-1 task, **TagPR**'s accuracy of 0.803 surpasses both Gemini-2.5-Pro (0.761) and GPT-4o (0.733). This trend of a much smaller model achieving superior performance is observed across the entire benchmark.

## 4.3 ABLATION STUDY

To dissect the contribution of each component within our framework, we conducted a comprehensive ablation study, with results presented in Table 1. Our analysis reveals a strong synergy, wherein each module proves indispensable for achieving the final performance.

The results first highlight the critical roles of the foundational training stages. The initial **SFT phase** is essential for bootstrapping the model with our tagged reasoning syntax. Its removal (TagPR w/o SFT) causes a significant performance drop (e.g., LaMP-1 accuracy falls from 0.803 to 0.747), demonstrating its necessity in establishing a structured thought process. Building upon this, the multi-stage **RL process** is vital for refining this structure into high-quality, personalized logic. The SFT-only model (TagPR w/o RL) exhibits a substantial performance gap (e.g., MAE on LaMP-3 increases from 0.218 to 0.339), underscoring RL's importance in refining the model's capacity for personalization reasoning.

Furthermore, our novel reward signals are proven to be highly effective. The **PRMU reward** provides a crucial user-aware signal. Its removal (TagPR w/o Reward) leads to a decline across all tasks. Crucially, the **tag-based reward** makes a substantial contribution by enforcing a logically coherent thought process. Its exclusion (TagPR w/o Tag) results in a sharp performance degradation (e.g., LaMP-2 F1-score drops from 0.557 to 0.511). Finally, our **two-stage training design** is validated as superior to a single, continuous RL stage. The initial guided phase instills robust reasoning, while the subsequent exploratory phase allows the model to refine its policy and achieve peak results. Collectively, these findings affirm that the synergistic integration of each carefully designed component is the key to TagPR's success.

## 4.4 Generalization Assessment

To evaluate whether **TagPR** learns a transferable personalization skill, rather than overfitting to the training distribution, we assess its zero-shot generalization performance on a new benchmark. We constructed this benchmark from Dianping[2], a prominent Chinese user-generated content platform. This setup poses a stringent test involving unseen domains, task formats, and a different language.

The benchmark consists of three distinct tasks derived from the post histories of 1,000 users. The tasks are: **(1) Dianping-Content**, generating post content from a title; **(2) Dianping-Title**, the inverse task of generating a title from content; and **(3) Dianping-Paraph**, rewriting a generic post to match a user's unique writing style. More detailed benchmark introduction is provided in the Appendix H.

Table 2: Zero-shot cross-lingual generalization performance on the three Dianping datasets. The best results are in **bold**, and the second-best are underlined. Our TagPR demonstrates superior performance.

| Dataset → | Dianping-Content | | Dianping-Title | | Dianping-Paraph | |
|---|---|---|---|---|---|---|
| Method | R-1 ↑ | R-L ↑ | R-1 ↑ | R-L ↑ | R-1 ↑ | R-L ↑ |
| RAG | 0.200 | 0.151 | 0.209 | 0.184 | 0.598 | 0.568 |
| RAG-R (Base) | 0.183 | 0.144 | 0.197 | 0.173 | 0.517 | 0.461 |
| SFT | 0.189 | 0.123 | 0.228 | 0.210 | 0.603 | 0.571 |
| SFT-R | 0.187 | 0.145 | 0.198 | 0.177 | 0.498 | 0.423 |
| GPT-4o | 0.207 | 0.168 | 0.236 | 0.211 | 0.606 | 0.573 |
| Gemini-2.5-Pro | **0.217** | 0.170 | 0.215 | 0.195 | 0.564 | 0.475 |
| **TagPR** | 0.216 | **0.171** | **0.240** | **0.218** | **0.617** | **0.583** |

As shown in Table 2, **TagPR** demonstrates exceptional generalization capabilities. It achieves state-of-the-art results across the benchmark, securing the top score on the majority of metrics and outperforming SFT method, which performs poor generalization, and leading proprietary models like GPT-4o. Our "tagging the thought" method, **TagPR**, creates a highly generalizable personalization reasoning model effective across diverse domains, tasks, and languages.

To further assess the generalization capabilities of TagPR, we evaluated the model on partial test sets of LongLaMP (Kumar et al., 2024) and PGraphRAG (Au et al., 2025) datasets. These benchmarks are characterized by ultra-long contexts and sparse user history profiles, respectively (detailed configurations are provided in the Appendix D). As shown in Table 3, our model demonstrated superior performance, verifying the robust transferability of its learned personalized reasoning capability. Furthermore, building upon the SFT-trained model and collected tags, we conducted retraining on the LongLaMP dataset employing only our proposed two-stage RL stage. The results (Table 7 in Appendix D) confirm that our method achieves excellent performance via direct RL optimization, eliminating the need for massive data annotation in the first stage.

---

[2]https://www.dianping.com/.

Table 3: Zero-shot generalization performance on partial test sets of LongLaMP and PGraphRAG. We report ROUGE-1 (R-1) and ROUGE-L (R-L) scores. The best results are in **bold**. Our TagPR demonstrates superior performance.

| Dataset → | TopicWriting | | ProductReview | | AbstractGeneration | | AmazonReviewTitle | |
|---|---|---|---|---|---|---|---|---|
| Method | R-1 ↑ | R-L ↑ | R-1 ↑ | R-L ↑ | R-1 ↑ | R-L ↑ | R-1 ↑ | R-L ↑ |
| Qwen3-8B-Instruct | 0.282 | 0.134 | 0.342 | 0.154 | 0.382 | 0.203 | 0.172 | 0.163 |
| Qwen3-8B-Thinking (Base) | 0.271 | 0.124 | 0.321 | 0.150 | 0.351 | 0.182 | 0.179 | 0.165 |
| Qwen3-32B-Instruct | 0.292 | 0.134 | 0.354 | 0.159 | 0.384 | 0.202 | 0.171 | 0.161 |
| Qwen3-32B-Thinking | 0.275 | 0.119 | 0.320 | 0.144 | 0.346 | 0.177 | 0.198 | 0.191 |
| GPT-4o | 0.294 | 0.140 | 0.330 | 0.157 | 0.372 | 0.200 | 0.140 | 0.136 |
| Gemini-2.5-Pro | 0.270 | 0.132 | 0.372 | 0.171 | 0.398 | 0.215 | 0.204 | 0.196 |
| TagPR (Our) | **0.303** | **0.154** | **0.377** | **0.175** | **0.404** | **0.221** | **0.210** | **0.202** |

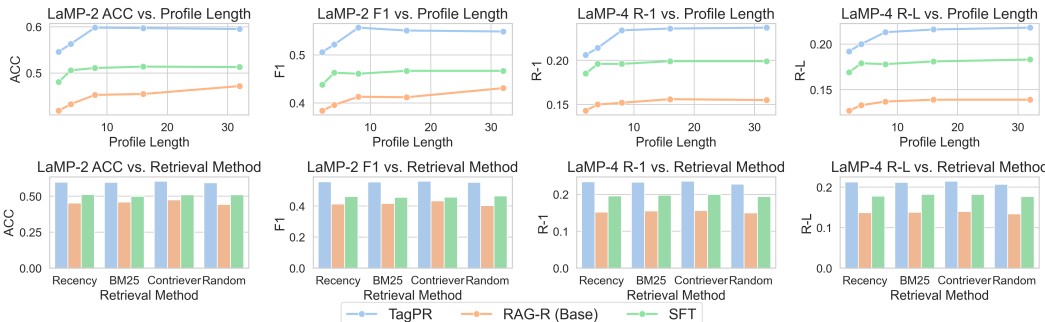

Figure 4: Robustness assessment of TagPR on LaMP-2 and LaMP-4. **Top:** Performance across varying profile lengths. **Bottom:** Performance across different retrieval methods. TagPR consistently outperforms baselines, demonstrating high data efficiency and resilience to retrieval quality.

## 4.5 ROBUSTNESS ASSESSMENT

We evaluate the robustness of **TagPR** against baselines SFT and Base by varying two key factors: user profile length and profile retrieval method. Figure 4 presents the results on the representative LaMP-2 and LaMP-4 tasks, with complete results available in the Appendix E.

First, we analyze the effect of profile length by varying the number of historical interactions from 2 to 32. The top row of Figure 4 shows that **TagPR** consistently outperforms the baselines across all lengths. Notably, TagPR's performance improves rapidly and starts to plateau with just 8 interactions, indicating its high data efficiency in distilling user preferences. In contrast, the baselines show more gradual improvements and maintain a significant performance gap.

Second, we assess the model's sensitivity to the profile retrieval method. We compare our default Recency-based retriever with three alternatives: a sparse retriever (BM25), a dense retriever (Contriever), and Random selection. As shown in the bottom row, **TagPR** demonstrates remarkable stability and maintains its superior performance across all retrieval strategies. Even with randomly selected profiles, TagPR's performance degradation is minimal, suggesting its reasoning process can effectively identify and utilize relevant information regardless of the profile quality.

## 4.6 FURTHER ANALYSIS

This section validates the PRMU design and analyzes length and tags distribution of the tagged reasoning chains, with further case studies and reasoning content analysis available in the Appendix B.

### 4.6.1 PERSONALIZATION REWARD MODEL DESIGN

To validate our proposed PRMU, we conducted a comprehensive ablation study to assess the contribution of its core components. The results, detailed in Table 4, compare four configurations: our full PRMU, PRMU without user embeddings (w/o UE), a baseline using an untrained reward model

Table 4: Ablation study of PRMU components across LaMP benchmarks.

| Dataset → | LaMP-1 | | LaMP-2 | | LaMP-3 | | LaMP-4 | | LaMP-5 | | LaMP-7 | |
|---|---|---|---|---|---|---|---|---|---|---|---|---|
| Method | ACC↑ | F1↑ | ACC↑ | F1↑ | MAE↓ | RMSE↓ | R-1↑ | R-L↑ | R-1↑ | R-L↑ | R-1↑ | R-L↑ |
| w/o RM | 0.768 | 0.769 | 0.557 | 0.514 | 0.246 | 0.393 | 0.205 | 0.190 | 0.522 | 0.453 | 0.545 | 0.490 |
| Untrained RM | 0.771 | 0.772 | 0.533 | 0.495 | 0.246 | 0.361 | 0.207 | 0.195 | 0.536 | 0.459 | 0.545 | 0.487 |
| PRMU w/o UE | 0.784 | 0.784 | 0.581 | 0.541 | 0.231 | 0.299 | 0.215 | 0.197 | 0.536 | 0.467 | 0.558 | 0.501 |
| PRMU | **0.803** | **0.803** | **0.598** | **0.557** | **0.218** | **0.263** | **0.234** | **0.213** | **0.542** | **0.471** | **0.565** | **0.507** |

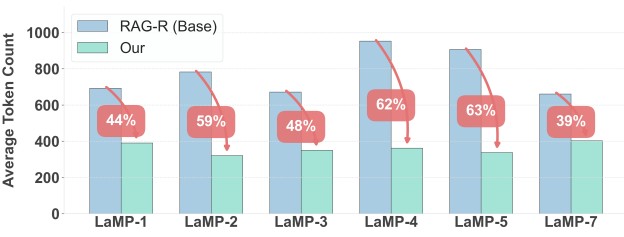 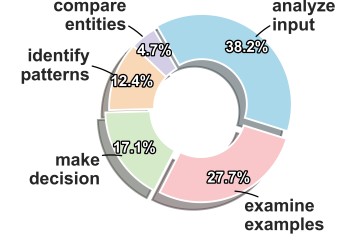

Figure 5: **Left**: Comparison of reasoning chain length between TagPR and Base on the LaMP validation set. **Right**: Frequency distribution of the five core reasoning tags generated by our model.

(Untrained RM), and a baseline with no reward model (w/o RM). Our findings first reveal that employing an off-the-shelf reward model offers no consistent advantage over having no reward model at all. In fact, it proved detrimental in certain cases (e.g., LaMP-2 F1 score), yielding a noisy and misaligned signal. Next, training the reward model on our personalization dataset, even without user-specific information (PRMU w/o UE), yields substantial improvements across all metrics. The most significant performance gains, however, are realized with the full PRMU model. By integrating user embeddings to provide a user-aware reward, PRMU consistently outperforms all other variants.

### 4.6.2 Tagged Reasoning Chains Analysis

**Reasoning Length** To assess reasoning efficiency, we compare the average token count of reasoning chains generated by our trained model against the original Qwen3-8B (Base) on the LaMP validation set. As illustrated in Figure 5 (Left), **TagPR** consistently produces more concise reasoning chains, achieving an average token reduction of over 50%. While the Base often generates verbose explorations, our "tagging the thought" framework guides the model along a direct logical path, effectively pruning irrelevant steps.

**Reasoning Tags** As shown in Figure 5 (Right), the distribution of reasoning tags reveals a structured cognitive process. The model prioritizes evidence gathering by heavily relying on `<analyze_input>` (38.2%) and `<examine_examples>` (27.7%). Subsequently, it performs higher-level synthesis and decision-making through `<identify_patterns>` (12.4%), `<compare_entities>` (4.7%), and `<make_decision>` (17.1%). This logical sequence confirms a coherent flow from analysis to personalized decision.

## 5 Conclusion

In this work, we introduce **TagPR**, a novel training framework that fundamentally enhances the personalization reasoning capabilities of LLMs. Our method first uses a data-driven pipeline to automatically create a dataset of tagged reasoning chains. We then employ a synergistic training strategy, combining SFT with a multi-stage RL process guided by a novel Personalization Reward Model with User Embeddings (PRMU). Extensive experiments show our approach achieves state-of-the-art results on the LaMP benchmark, outperforming even large proprietary models and demonstrating strong generalization. This work validates that training LLMs to generate structured, interpretable reasoning is a highly effective pathway to unlocking genuine personalization, paving the way for more sophisticated and user-aligned intelligent systems.

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

## A  DETAILED EXPERIMENTAL SETUP

This section provides a detailed description of our experimental setup, including implementation details, benchmark information, and baseline configurations.

### A.1  IMPLEMENTATION DETAILS

**Backbone Model** We use Qwen3-8B (Team, 2025) as our base model for all experiments unless otherwise specified.

**Supervised Fine-Tuning (SFT)** The SFT stage was conducted on 8 A100 GPUs. We used a learning rate of 1e-5 and a global batch size of 64. The model was trained for 2 epochs on the dataset described in Section 3.2.

**Reinforcement Learning (RL)** Data Sampling: We sampled data from the LaMP training set for RL. Specifically, we randomly sampled 1,024 examples for each of the LaMP-1, LaMP-3, LaMP-4, LaMP-5, and LaMP-7 tasks. For the more challenging LaMP-2 task, we sampled 3,200 examples. Training Parameters: The first RL stage was trained for 13 epochs, and the second stage was trained for 2 epochs. Both stages were conducted on 8 A100 GPUs with a global batch size of 128 and a learning rate of 1e-6. Policy Rollout: During the policy rollout stage, we set the temperature to 1.0 and top-p to 1.0, generating 5 responses for each prompt. Other Hyperparameters: The low and high clip ratios for the GSPO algorithm were set to 0.0003 and 0.0004, respectively. For the repetition penalty reward, we used n-grams of size 4. For the tag reward, the minimum required number of tags was set to 3.

### A.2  BENCHMARK DETAILS

**Dataset** We use the LaMP benchmark, a widely-adopted benchmark for evaluating the personalization capabilities of LLMs. It requires models to analyze user historical profiles to answer current queries. Since the official test set is not publicly available, all our evaluations are conducted on the official validation set. LaMP-6 was excluded from our evaluation due to its unavailability. We evaluated on the complete validation dataset for all other tasks. The detailed data statistics of LaMP is shown in Table 5

**Evaluation Metrics** Following the original LaMP benchmark, we employ the following metrics:

Table 5: Data statistics of the LaMP benchmark.

| Task | Task Type | #Train | #Val | #Classes |
|---|---|---|---|---|
| LaMP-1 | Binary classification | 6,542 | 1,500 | 2 |
| LaMP-2 | Categorical classification | 5,073 | 1,410 | 15 |
| LaMP-3 | Ordinal classification | 20,000 | 2,500 | 5 |
| LaMP-4 | Text generation | 12,500 | 1,500 | - |
| LaMP-5 | Text generation | 14,682 | 1,500 | - |
| LaMP-7 | Text generation | 13,437 | 1,498 | - |

**Classification Tasks (LaMP-1 & LaMP-2).** Task Descriptions: LaMP-1 (Personalized Citation): A binary classification task where the model determines which of two candidate references an author is likely to cite based on their profile. LaMP-2 (Personalized Movie Tagging): A multi-class classification task requiring the model to select the most appropriate tag from 15 candidates for a movie, conditioned on the user's tagging history. Metrics & Calculation: We report Accuracy and F1-Score. For Accuracy, we calculate the ratio of correctly predicted samples to the total samples. For F1-Score: In LaMP-1 (binary), we report the standard F1 for the positive class. In LaMP-2 (multi-class), we follow the benchmark standard and report Macro-F1.

**Rating Task (LaMP-3).** Task Description: LaMP-3 (Personalized Product Rating) is an ordinal classification task. The model predicts a specific integer rating (1–5 stars) for a product based on the user's historical reviews. Metrics: We employ MAE (Mean Absolute Error) and RMSE (Root Mean Square Error). These metrics are standard for quantifying the deviation between predicted scores and ground-truth user ratings.

**Text Generation Tasks (LaMP-4, 5, 7).** Task Descriptions: These tasks involve generating text tailored to a user's specific style: LaMP-4: Personalized news headline generation. LaMP-5: Personalized scholarly title generation. LaMP-7: Personalized tweet paraphrasing. Metrics: We utilize ROUGE-1 and ROUGE-L. ROUGE-1: measures the overlap of unigrams to assess content adequacy. ROUGE-L: is based on the Longest Common Subsequence, evaluating fluency and structural similarity between the generated text and the reference.

### A.3 BASELINES AND COMPARISON SETUP

To rigorously evaluate our proposed method, we benchmark it against a wide spectrum of baselines. For a fair comparison, all methods are built upon the Qwen3-8B base model and utilize the user's 8 most recent profiles as input, unless specified otherwise (e.g., proprietary models like GPT-4o).

The baselines are categorized as follows. Standard methodologies include Zero-shot, which generates responses without user profiles as a non-personalized lower bound; standard Retrieval-Augmented Generation (RAG); Personalization-Augmented Generation (PAG) (Richardson et al., 2023), which enhances RAG with user history summaries; Supervised Fine-Tuning (SFT) on the full dataset; and SFT-Ind, which is fine-tuned only on individual task data. Reasoning-enhanced variants of these methods, denoted with a '-R' suffix, are also included. We further compare against advanced techniques. Personalization-focused methods include PPlug (Liu et al., 2024), a plug-and-play approach using specialized user embeddings, and HYDRA-Adapter (Zhuang et al., 2024), for which we use only its adapter version to maintain a consistent retrieval method for fairness. Reasoning-focused baselines include R2P (Luo et al., 2025b), which employs hierarchical reasoning templates, and PrLM (Zhang et al., 2025), which uses a contrastive reward model with reinforcement learning. To situate our method's performance against the frontier of language models, we also include several leading state-of-the-art LLMs: GPT-4o (Hurst et al., 2024), Gemini-2.5-Pro (Comanici et al., 2025), Qwen3-235B-A22B (Team, 2025), and Deepseek-R1 (Guo et al., 2025).

We deliberately exclude methods centered on optimizing the retrieval module, as improving retrieval is not the focus of our research. Additionally, we do not compare against OPPU (Tan et al., 2024b), as its approach requires fine-tuning a unique module for every user and presupposes the availability of extensive user-specific profiles, rendering it infeasible to implement across the full validation set.

# B  ADDITIONAL FURTHER ANALYSIS

## B.1  CASE STUDY

We present a qualitative case study from the LaMP-2 benchmark to illustrate the advanced personalization reasoning of our proposed **TagPR** in Figure 1. The task is to assign a suitable tag to a movie based on a user's interaction history.

The baseline model, Qwen3-8B, exhibits a generic reasoning approach, focusing exclusively on the semantics of the new item's description. For instance, it interprets the phrase "bankrupt USA" as a form of social critique, subsequently outputting the tag social commentary. While this inference is plausible in isolation, it disregards the user's distinct historical preferences, resulting in a generic and incorrect recommendation.

In stark contrast, **TagPR** demonstrates a structured, user-centric reasoning process. Its chain-of-thought explicitly follows a sequence of operational steps demarcated by tags: `<examine_examples>`, `<identify_patterns>`, and `<compare_entities>`. The model first analyzes the user's profile to discern their specific conceptualization of "dystopia" from historical data. It then aligns the new movie with this inferred user-specific logic, correctly concluding that the narrative fits the established pattern. Consequently, **TagPR** produces the correct tag: "dystopia".

This comparative analysis highlights that **TagPR** transcends generic semantic interpretation to effectively model and apply a user's unique reasoning patterns. This capability constitutes a more authentic form of personalization reasoning, a task at which the baseline model fails.

Additional comparative cases are presented below.

---

**Response from Qwen3-8B for Case 1**

<think>
Okay, let's see. The user is asking which tag applies to the movie description they provided. The tags to choose from are: ...
The movie description is about a 17-year-old named Mariah whose parents vanish and his brother is kidnapped. He follows clues to a hotel, discovers a hidden realm with child-stealing monsters, deadly secrets, and a magical artifact called the Midas Box. The story involves a curse with supernatural power and the stakes of saving his world and family.
First, I need to check which of the given tags fit. Let's go through them one by one.
Sci-fi: ...
Based on a book: ...
Comedy: ...
Action: ...
...
True story: ...
So, considering all that, the most fitting tags would be "fantasy" due to the supernatural elements and hidden realm. Also, "action" because of the adventure and risk involved. However, the user's previous examples had "based on a book" for some movies, but this one doesn't mention that. The twist ending was used in another example, but the current description doesn't explicitly state a twist. However, the presence of a hidden realm and a curse might imply a twist. But the main elements are fantasy and action. Wait, the user's previous examples had "based on a book" for several movies, but the current description doesn't mention that. So maybe the answer is "fantasy" or "action". But looking back, the user's examples had "based on a book" for movies like "An Italian diplomat's son..." and others, but the current movie's description doesn't mention being based on a book. So "based on a book" might not be the right tag here. The key elements are supernatural, hidden realm, monsters, artifact, which are fantasy elements. So the answer is "fantasy".
</think>
fantasy

---

**Response from TagPR for Case 1**

<think>
<analyze_input>We are given several examples of movie descriptions and their corresponding tags. Then we are given a new movie description and asked to choose the appropriate tag from a list. </analyze_input>
<examine_examples>Let's look at the examples: 1. "An oppressed Mexican peasant village hires seven gunfighters to help defend their homes." → tag: "twist ending"

...

8. "Harvard symbologist Robert Langdon is recruited by the Vatican..." → tag: "based on a book" </examine_examples>
<identify_patterns>We notice that all the examples that are tagged "based on a book" are movies that are adaptations of books. Specifically:

- The first example is "The Magnificent Seven" (1960), which is based on

    ...

- The eighth example is "Angels & Demons" (2009), based on a novel by Dan Brown.

</identify_patterns>
<analyze_input>Now, let's look at the new movie description: "Seventeen-year-old Mariah Mundi's life is turned upside down when his parents vanish and his younger brother is kidnapped. Following a trail of clues to the darkly majestic Prince Regent Hotel, Mariah discovers a hidden realm of child-stealing monsters, deadly secrets and a long-lost artefact that grants limitless wealth—but also devastating supernatural power. With the fate of his world, and his family at stake, Mariah will risk everything to unravel the Curse of the Midas Box." </analyze_input>
<examine_examples>This description is for the movie "The Midas Touch" (2003), which is based on the novel "The Curse of the Midas Box" by John C. Wright. </examine_examples>
<identify_patterns>Therefore, the appropriate tag for this movie is "based on a book". </identify_patterns>
<make_decision>Let's check the list of tags provided: [sci-fi, based on a book, comedy, action, twist ending, dystopia, dark comedy, classic, psychology, fantasy, romance, thought-provoking, social commentary, violence, true story] </make_decision>
<verify_conclusion>We can see that "based on a book" is one of the tags. </verify_conclusion>
<formulate_conclusion>Therefore, the answer is "based on a book". </formulate_conclusion>
</think>
based on a book

**Response from Qwen3-8B for Case 2**

<think>

Okay, let's see. The user provided a list of papers by an author and is asking which of the two given references is related. The options are [1] "Compressed Linear Algebra for Large-Scale Machine Learning" and [2] "Database and information-retrieval methods for knowledge discovery".

First, I need to look at the titles of the author's papers to see which topic they're focusing on. The author's papers include things like Bayesian specification learning for API errors, distributed Bayesian machine learning, complex object implementations for big data, distributed algorithms for covariance matrices, grading in MOOCs, learning to grade student programs, comparing machine learning platforms, simulating database-valued Markov chains, and a model-agnostic framework for spatial anomaly detection.

Looking at these, several themes emerge: ...

Now, the two options are about compressed linear algebra for machine learning and database/information retrieval methods for knowledge discovery. ... Wait, the user's question is: "which reference is related?" given the list of the author's papers and the two options. So the user is asking which of the two options is related to the author's work.

So, the author's papers include topics like distributed Bayesian machine learning, big data systems, database simulations (SimSQL), and machine learning platforms. The first option, [1], is about compressed linear algebra for large-scale ML, which is a machine learning technique. The second option, [2], is about database and information retrieval methods for knowledge discovery. ... But the user is asking which of the two references is related. The answer is likely [2], but I'm not 100% certain. Alternatively, maybe [1] is related because the author's work on large-scale machine learning (like the paper on distributed algorithms) could be connected to compressed linear algebra. But the user's options are two references, and the answer is to choose one. Given that the author has a paper on SimSQL (database-related), [2] is about databases and information retrieval, so [2] is more directly related. Therefore, the answer is [2].

</think>

[2]

**Response from TagPR for Case 2**

<think>
<analyze_input>
Okay, let me try to figure out which reference is related to the author's papers. The user provided a list of papers the author has written and two options for references: [1] "Compressed Linear Algebra for Large-Scale Machine Learning" and [2] "Database and information-retrieval methods for knowledge discovery".
</analyze_input>
<examine_examples>
First, I need to look at the titles of the author's papers to see what themes they're working on. Let me list them out again:
1. "Bayesian specification learning for finding API usage errors"
...
9. "A Model-Agnostic Framework for Fast Spatial Anomaly Detection"
</examine_examples>
<identify_patterns>
Looking at these, several themes emerge. There's a strong focus on machine learning, especially distributed and large-scale aspects. Papers like "The BUDS Language for Distributed Bayesian Machine Learning" and "A comparison of platforms for implementing and running very large scale machine learning algorithms" clearly relate to machine learning systems. Also, "Distributed Algorithms for Computing Very Large Thresholded Covariance Matrices" and "Compressed Linear Algebra for Large-Scale Machine Learning" (which is reference [1]) might be connected because they both deal with large-scale computations in machine learning.
</identify_patterns>
<evaluate_reference>
Then there's the database-related paper: ...
</evaluate_reference>
<compare_entities>
Now, comparing the two references: [1] is about compressed linear algebra for machine learning, which aligns with papers on large-scale ML algorithms and distributed computing. [2] is about database and information-retrieval methods for knowledge discovery, which connects to the database-valued Markov chains paper.
</compare_entities>
<synthesize_findings>
But the author has multiple papers on machine learning, especially distributed and large-scale. ...
</synthesize_findings>
<make_decision>
So, reference [1] seems more broadly related to the author's work in machine learning systems and large-scale algorithms. Reference [2] is only tangentially related through one paper, while [1] connects to multiple papers.
</make_decision>
</think>
[1]

**Response from Qwen3-8B for Case 3**

<think>
Okay, let's see. The user wants me to paraphrase the tweet "@Ylsoccermomof3, I appreciate it and I'll enjoy my double cheeseburger." without any explanation before or after.

First, I need to understand the original message. The person is thanking @Ylsoccermomof3 and mentioning they'll enjoy a double cheeseburger. The key elements are gratitude and the mention of the food.

Now, paraphrasing means rewording it while keeping the same meaning. Let me think of synonyms. "Appreciate" could be "thank" or "grateful for." "Enjoy" might be "savor" or "look forward to." "Double cheeseburger" is specific, so maybe keep it as is or use "big cheeseburger" for a slight variation.

Also, the structure: the original starts with the handle, then the message. The user's example response uses "Thanks, @Ylsoccermomof3, and I'll savor my double cheeseburger." So they changed "appreciate" to "Thanks" and "enjoy" to "savor." That's a good approach.

I should check if there are other ways to phrase it. Maybe "I'm grateful, @Ylsoccermomof3, and I'll relish my double cheeseburger." But "relish" might be a bit more formal. The original is casual, so "savor" fits better.

Another option: "Thanks, @Ylsoccermomof3, and I'll enjoy my big cheeseburger." But "big" might not be the same as "double." The user's example used "double," so maybe stick with that.

I think the user's example is good, but maybe check for any other possible variations. However, the user wants a paraphrase, not necessarily the same as the example. Let me make sure the structure is correct. The handle is included, then the message.

...

Thanks, @Ylsoccermomof3, and I'll savor my double cheeseburger.
</think>

**Response from TagPR for Case 3**

<think>
<analyze_input>
Okay, the user has given me a task to paraphrase a specific tweet based on patterns observed from several example tweets they provided. Let me break this down carefully.
</analyze_input>
<examine_examples>
First, I need to analyze the patterns in the example tweets they shared. Looking at these:
- They all start with "@username" followed by a casual greeting or statement
- The language is very conversational with contractions ("you're", "it's")
- There's a mix of personal updates ("my cat is having kittens") and light-hearted interactions
- Some include multiple @mentions
- The tone is friendly and informal
</examine_examples>
<identify_patterns>
The key pattern I notice is the casual, conversational style with:
- Use of "I" statements
- Simple sentence structures
- No formal punctuation beyond basic commas
- Often includes personal context ("I just woke up", "I think everyone has")
</identify_patterns>
<analyze_input>
Now for the tweet to paraphrase: "@Ylsoccermomof3, I appreciate it and I'll enjoy my double cheeseburger."
</analyze_input>
<identify_patterns>
Hmm... the original has:
- Formal "I'll" instead of "I'm gonna"
- "double cheeseburger" which is slightly more specific than typical casual phrasing
- No contractions in "I'll" (though it's common in speech)
</identify_patterns>

...

<verify_conclusion>
So "I'm gonna" is better than "I'll" for that casual vibe. And "enjoy" could become "devour" like in the @Jintanut example ("very hungry") but "enjoy" is already good.
</verify_conclusion>
<formulate_conclusion>
Final decision: Keep the structure but make it sound like a real person talking. Add "thanks" to match the gratitude pattern, use "gonna" for casualness, and keep the @mention.
</formulate_conclusion>
</think>
@Ylsoccermomof3, thanks for the encouragement and I'm gonna enjoy my double cheeseburger.

## B.2 REASONING CONTENT ANALYSIS

To further investigate the reasoning processes qualitatively, we generated word clouds from the reasoning chains produced by the baseline Qwen3-8B and our **TagPR** model on the LaMP validation set, as shown in Figure 6. The visualization reveals a stark contrast in their reasoning styles.

The word cloud for the baseline model is populated by general, conversational terms such as "maybe", "think", "example", and "review". This indicates a descriptive, narrative-style reasoning process, where the model verbalizes a general thought process rather than executing a structured plan. In sharp contrast, the **TagPR** word cloud prominently features action-oriented keywords like

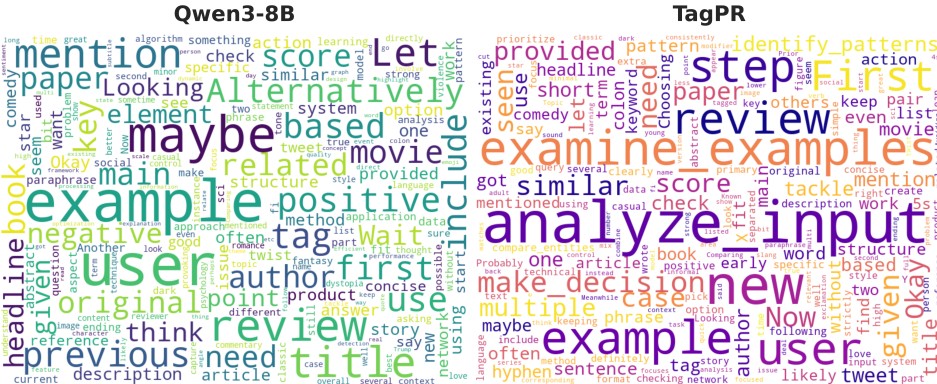

Figure 6: Word cloud comparison of reasoning chains from the baseline Qwen3-8B (left) and our TagPR model (right) on the LaMP validation set. TagPR's reasoning is dominated by action-oriented keywords derived from our functional tags.

"examine_examples", "analyze_input", "identify_patterns", and "make_decision", which are the core components of the functional tags introduced in our framework. This shift demonstrates that **TagPR** successfully learns to adopt an explicit, structured, and interpretable reasoning schema. Instead of merely describing its thought process, the model actively executes a sequence of defined logical steps, confirming a more efficient and targeted approach to personalization reasoning.

## C  HYPERPARAMETER SENSITIVITY ANALYSIS OF REWARD FUNCTION

We conducted a sensitivity analysis on the reward function hyperparameters by varying the weights of $\alpha$, $\beta$, and $\gamma$ while keeping all other experimental conditions constant. The results, presented in Table 6, indicate that the model remains robust within a reasonable range of parameter variations, with our current configuration yielding the most balanced performance.

Table 6: Hyperparameters sensitivity analysis of reward on LaMP. The best results are in **bold**.

| Dataset → | LaMP-1 | | LaMP-2 | | LaMP-3 | | LaMP-4 | | LaMP-5 | | LaMP-7 | |
|---|---|---|---|---|---|---|---|---|---|---|---|---|
| Method | ACC ↑ | F1 ↑ | ACC ↑ | F1 ↑ | MAE ↓ | RMSE ↓ | R-1 ↑ | R-L ↑ | R-1 ↑ | R-L ↑ | R-1 ↑ | R-L ↑ |
| $\alpha = 1.0, \beta = 1.0, \gamma = 1.0$ | 0.797 | 0.797 | 0.580 | 0.552 | 0.226 | 0.271 | 0.226 | 0.208 | 0.540 | 0.469 | 0.563 | 0.504 |
| $\alpha = 0.2, \beta = 0.2, \gamma = 0.8$ | 0.789 | 0.788 | 0.572 | 0.531 | 0.230 | 0.272 | 0.225 | 0.207 | 0.532 | 0.459 | 0.561 | 0.500 |
| $\alpha = 0.8, \beta = 0.2, \gamma = 0.2$ | 0.789 | 0.789 | 0.575 | 0.536 | 0.229 | 0.272 | 0.227 | 0.208 | 0.539 | 0.466 | 0.564 | 0.505 |
| $\alpha = 0.2, \beta = 0.8, \gamma = 0.2$ | 0.785 | 0.785 | 0.573 | 0.535 | 0.235 | 0.275 | 0.225 | 0.206 | 0.534 | 0.459 | 0.560 | 0.498 |
| $\alpha = 0.8, \beta = 0.8, \gamma = 0.4$ | 0.801 | 0.800 | 0.594 | 0.553 | **0.216** | **0.262** | 0.231 | 0.212 | **0.542** | 0.470 | 0.564 | 0.505 |
| $\alpha = 0.8, \beta = 0.8, \gamma = 0.2$ (Our) | **0.803** | **0.803** | **0.598** | **0.557** | 0.218 | 0.263 | **0.234** | **0.213** | **0.542** | **0.471** | **0.565** | **0.507** |

## D  SUPPLEMENT FOR GENERALIZATION ASSESSMENT

### D.1  GENERALIZATION ASSESSMENT ON LONGLAMP

We further evaluated the generalization capabilities of our model using the LongLaMP dataset. Characterized by extensive user history profiles and a requirement for long-form text generation, LongLaMP poses a significant challenge to personalized reasoning. We focused on three tasks: TopicWriting, ProductReview, and AbstractReviewTitle. Due to computational constraints, we randomly sampled 10% of the test set for each task and reported performance using ROUGE-1 (R-1) and ROUGE-L (R-L).

Zero-shot Performance: We first assessed zero-shot generalization by directly applying the model trained on the LaMP dataset to the LongLaMP dataset. As shown in Table 3, our model demonstrated robust zero-shot generalization capabilities.

Transfer Learning Capability: Subsequently, we evaluated the transferability of our proposed method. In this experiment, we bypassed the construction of tagged reasoning chains and the initial

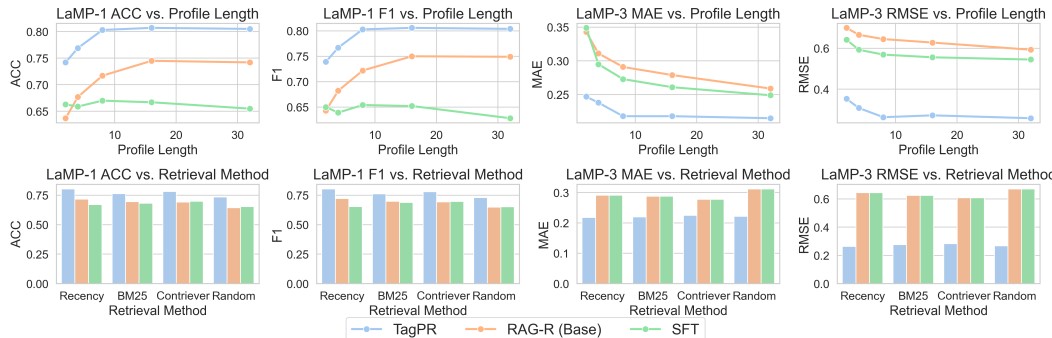

Figure 7: Robustness assessment of TagPR on LaMP-1 and LaMP-3. **Top:** Performance across varying profile lengths. **Bottom:** Performance across different retrieval methods.

Supervised SFT on the target data. Instead, we initialized the model using the LaMP-SFT checkpoint and collected tags, then performed two-stage RL directly on the LongLaMP training set. The RL training data consisted of 3,072 samples, with 1,024 instances randomly sampled from each of the three tasks. The results in Table 7 illustrate strong performance following transfer training, validating the effectiveness of our approach.

Table 7: Transfer Learning performance on partial test sets of LongLaMP. We report ROUGE-1 (R-1) and ROUGE-L (R-L) scores. The best results are in **bold**.

| Dataset → | TopicWriting | | ProductReview | | AbstractGeneration | |
|---|---|---|---|---|---|---|
| **Method** | **R-1 ↑** | **R-L ↑** | **R-1 ↑** | **R-L ↑** | **R-1 ↑** | **R-L ↑** |
| **GPT-4o** | 0.294 | 0.140 | 0.330 | 0.157 | 0.372 | 0.200 |
| **Gemini-2.5-Pro** | 0.270 | 0.132 | 0.372 | 0.171 | 0.398 | 0.215 |
| **Qwen3-8B-Thinking (Base)** | 0.271 | 0.124 | 0.321 | 0.150 | 0.351 | 0.182 |
| **TagPR (Zero-shot Generalization)** | 0.303 | 0.154 | 0.377 | 0.175 | 0.404 | 0.221 |
| **TagPR (Trained on LongLaMP)** | **0.376** | **0.173** | **0.434** | **0.197** | **0.478** | **0.256** |

### D.2 GENERALIZATINO ASSESSMENT ON PGRAPHRAG

We further evaluated the model's generalization capabilities using the PGraphRAG dataset. This dataset is characterized by extreme sparsity, where the majority of user profiles contain either zero or only one, posing a significant challenge for personalized reasoning. We focused specifically on the AmazonReviewTitle subtask. Since historical profiles are the foundation of personalization, we restricted our evaluation to test users with a profile history length of at least one, excluding neighbor profiles information. By directly applying our trained model, we demonstrated its robust personalized reasoning capabilities even in data-sparse scenarios, as shown in Table 3.

## E SUPPLEMENT FOR ROBUSTNESS ASSESSMENT

This section provides supplementary results for the robustness assessment discussed in the main paper. Figure 7 and Figure 8 illustrates the performance of **TagPR** against the SFT and Base baselines on the LaMP-1, LaMP-3, LaMP-5, and LaMP-7 tasks, complementing the results for LaMP-2 and LaMP-4 shown in Figure 4.

As demonstrated in the figure, the conclusions from the main text hold true across these additional datasets. **TagPR** consistently achieves superior performance, showcasing high data efficiency by reaching a strong performance level with only a few user interactions. Furthermore, its advantage is maintained across all profile retrieval methods, including random selection, which underscores the robustness of our framework.

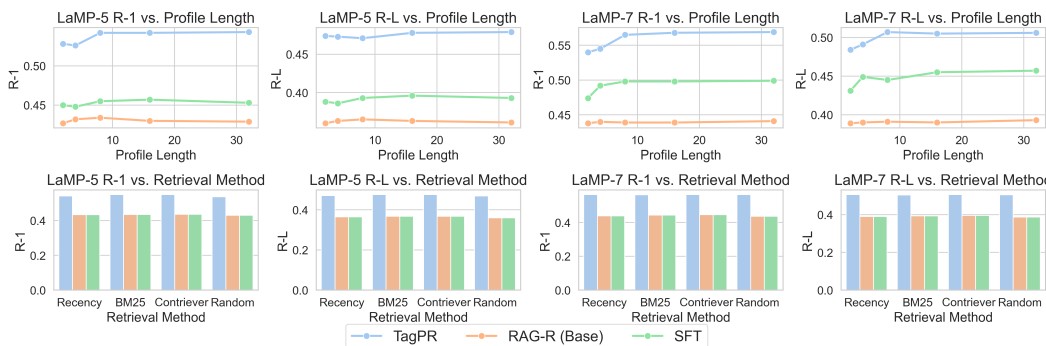

Figure 8: Robustness assessment of TagPR on LaMP-5 and LaMP-7. **Top:** Performance across varying profile lengths. **Bottom:** Performance across different retrieval methods.

# F    SUPPLEMENT FOR TAGGED REASONING CHAINS CONSTRUCTION

## F.1    PROMPTS DETAILS

### F.1.1    THE PROMPT FOR LLM FILTER

---

**Prompt for LLM Filter**

# Role
You are an AI expert specializing in evaluating the Chain-of-Thought (CoT) quality of large language models. Your task is to provide a comprehensive and objective evaluation of the model's Chain-of-Thought quality based on the provided question and the model's response.

# Task Description
I will provide you with a "Question" and a "Model Response" generated by a large language model for that question. The "Model Response" contains detailed reasoning steps and the final answer. Please evaluate the quality of the Chain-of-Thought in this "Model Response" according to the following evaluation dimensions, and strictly output the result in the specified JSON format.

# Evaluation Dimensions
1. **Logical Coherence:** Is there a clear logical connection between the reasoning steps? Are there any logical leaps or contradictions? (1-5 points)

2. **Step Accuracy:** Is every step in the reasoning chain accurate? Are there any factual errors or calculation mistakes? (1-5 points)

3. **Reasoning Completeness:** Does the Chain-of-Thought cover all the key steps required to solve the problem? Are there any omissions? (1-5 points)

4. **Relevance to the Question:** Is the entire thought process closely centered around the original question? Is there any redundant or off-topic reasoning? (1-5 points)

# Input Data
[Question]
{question}

[Model Response]
{response}

# Output Format

---

Please strictly follow the JSON format below for your evaluation results. Ensure the output is a complete and syntactically correct JSON object. Do not add any additional explanations or text before or after the JSON code block.

```json
{
  "evaluation_report": {
  "detailed_assessment": [
      {
        "dimension": "Logical Coherence",
        "reasoning": "[Provide an explanation of the pros and cons
            for this dimension]",
        "score": "[Enter an integer score from 1-5 here]"
      },
      {
        "dimension": "Step Accuracy",
        "reasoning": "[Provide an explanation of the pros and cons
            for this dimension, and explicitly point out any errors
            if they exist]",
        "score": "[Enter an integer score from 1-5 here]"
      },
      {
        "dimension": "Reasoning Completeness",
        "reasoning": "[Provide an explanation of the pros and cons
            for this dimension, and explicitly point out any
            omissions if they exist]",
        "score": "[Enter an integer score from 1-5 here]"
      },
      {
        "dimension": "Relevance to the Question",
        "reasoning": "[Provide an explanation of the pros and cons
            for this dimension, such as the presence of redundant
            information]",
        "score": "[Enter an integer score from 1-5 here]"
      }
    ],
    "summary": {
      "total_score": "[Enter the total score, between 1 and 20,
          which is the sum of the scores from each dimension]"
    }
  }
}
```

### F.1.2 THE PROMPT FOR EXPLORATORY TAGGING

**Prompt for Exploratory Tagging**

**Role:** You are an expert specializing in understanding and analyzing the thought processes of AI. Your task is to carefully review a given question and the "Chain-of-Thought" generated by an AI model to answer it. You will then break down this Chain-of-Thought into meaningful segments and assign an XML-style tag to each segment that best describes its function.

**Task:**
Based on the user-provided **[Question]** and the model-generated **[Chain-of-Thought]**, please complete the following steps:

1. **Analyze the Question and Chain-of-Thought:** Deeply understand the core requirements of the question and how the Chain-of-Thought progressively derives the final answer.

2. **Segment the Chain-of-Thought:** Break down the entire Chain-of-Thought into multiple logically coherent steps or stages. Each step should represent a distinct function in the thought process.

3. **Tag Each Segment:** For each segmented piece, create an XML-style tag (e.g., `<tag_name>...</tag_name>`). The tag name should accurately summarize the role this segment plays in the overall thought process.

4. **Output the Result:** Output only the tagged Chain-of-Thought text.

**Guidelines and Tag Suggestions:**

- **Tags should reflect function:** The tag name should describe "what this segment is doing", not simply restate its content.
- **Maintain consistency:** Try to use the same tags for similar functions.
- **Tags should be concise and clear:** Use short, easy-to-understand words for tag names (English is recommended for easier programmatic processing).

**Example:**
**[Question]:**
History: "When brilliant video game maker Flynn hacks the mainframe of his ex-employer..." → Tag: "classic" and, "When an insatiable great white shark terrorizes the townspeople..." → Tag: "classic" and, "Jake Blues, just released from prison, puts his old band back together..." → Tag: "classic" and, ...(other examples of "classic" and "true story")...
Description: "In the years before the Second World War, a tomboyish postulant at an Austrian abbey is hired as a governess in the home of a widowed naval captain with seven children, and brings a new love of life and music into the home."
Candidate Tags: [sci-fi, based on a book, comedy, action, ..., classic, ..., true story]

**[Chain-of-Thought to be tagged]:**
First, I analyzed the historical data provided by the user. This data shows descriptions of several movies and their corresponding tags.
I found that the movies tagged as "classic" (like *Tron*, *Jaws*, *The Blues Brothers*, etc.) are all well-known and beloved films released in the 20th century. Movies tagged as "true story" (like *Apollo 13*) are based on real events. This reveals a pattern: the "classic" tag points to older films with a significant era-defining and broad cultural impact.
Next, I analyzed the new movie description to be tagged: "In the years before the Second World War...". This description points to the famous movie *The Sound of Music*.
*The Sound of Music*, released in 1965, tells a household story and achieved immense success and cultural influence worldwide, perfectly matching the characteristics of a "classic" movie that I summarized from the historical data. It is not a sci-fi film, and while it might be partially based on real experiences, its core identity is its classic status.
Therefore, among all candidate tags, "classic" is the most appropriate.

**[Your Output]:**

```
<analyze_user_history>
First, I analyzed the historical data provided by the user. This
    data shows descriptions of several movies and their
    corresponding tags.
</analyze_user_history>
<identify_pattern>
I found that the movies tagged as "classic" (like *Tron*, *Jaws*, *
    The Blues Brothers*, etc.) are all well-known and beloved films
    released in the 20th century. Movies tagged as "true story" (
    like *Apollo 13*) are based on real events. This reveals a
    pattern: the "classic" tag points to older films with a
    significant era-defining and broad cultural impact.
</identify_pattern>
<analyze_current_query>
```

```
Next, I analyzed the new movie description to be tagged: "In the
    years before the Second World War...". This description points
    to the famous movie *The Sound of Music*.
</analyze_current_query>
<compare_query_with_history>
*The Sound of Music*, released in 1965, tells a household story and
    achieved immense success and cultural influence worldwide,
    perfectly matching the characteristics of a "classic" movie that
    I summarized from the historical data. It is not a sci-fi film,
    and while it might be partially based on real experiences, its
    core identity is its classic status.
</compare_query_with_history>
<final_conclusion>
Therefore, among all candidate tags, "classic" is the most
    appropriate.
</final_conclusion>
```

**Now, according to the rules above, please add tags to the [Question] and [Chain-of-Thought] provided below:**

**[Question]:**
{question}

**[Chain-of-Thought to be tagged]:**
{chain_of_thought}

### F.1.3   THE PROMPT FOR RESTRICTED TAGGING

---

**Prompt for Restricted Tagging**

**Role:** You are an expert specializing in understanding and analyzing the thought processes of AI. Your task is to carefully review a given question and the "Chain-of-Thought" generated by an AI model to answer it. You will then break down this Chain-of-Thought into meaningful segments and assign an XML-style tag to each segment that best describes its function.

**Task:**
Based on the user-provided **[Question]** and the model-generated **[Chain-of-Thought]**, please complete the following steps:

1. **Analyze the Question and Chain-of-Thought:** Deeply understand the core requirements of the question and how the Chain-of-Thought progressively derives the final answer.

2. **Segment the Chain-of-Thought:** Break down the entire Chain-of-Thought into multiple logically coherent steps or stages. Each step should represent a distinct function in the thought process.

3. **Tag Each Segment:** For each segmented piece, create an XML-style tag (e.g., `<tag_name>...</tag_name>`). The tag name must be chosen exclusively from the mandatory list provided below.

4. **Output the Result:** Output only the tagged Chain-of-Thought text.

**Mandatory Tag Set and Definitions:**
You **must** use **only** the tags from the following list. Choose the tag that best describes the function of each segment.

- **analyze_input**: Analyzes the initial user question or task description to understand the goal.

---

- **examine_examples**: Examines specific, individual pieces of evidence, data points, or examples provided.
- **identify_patterns**: Summarizes findings from one or more examples to find a common rule, pattern, or theme.
- **evaluate_reference**: Assesses how the input aligns with a specific, external piece of reference material.
- **compare_entities**: Performs a direct comparison between two or more items to determine their similarities, differences, or which is superior.
- **synthesize_findings**: Consolidates all prior analysis and comparisons into a comprehensive summary before making a final choice.
- **make_decision**: Commits to a specific, final choice or action.
- **verify_conclusion**: Performs a final check on the decision to ensure it is logical, consistent, and accurate.
- **formulate_conclusion**: Constructs the final, complete answer or statement based on the decision made.

**Example:**

**[Question]:**

History: "When brilliant video game maker Flynn hacks the mainframe of his ex-employer. . ." → Tag: "classic" and, "When an insatiable great white shark terrorizes the townspeople. . ." → Tag: "classic" and, "Jake Blues, just released from prison, puts his old band back together. . ." → Tag: "classic" and, . . . (other examples of "classic" and "true story"). . .

Description: "In the years before the Second World War, a tomboyish postulant at an Austrian abbey is hired as a governess in the home of a widowed naval captain with seven children, and brings a new love of life and music into the home."

Candidate Tags: [sci-fi, based on a book, comedy, action, . . . , classic, . . . , true story]

**[Chain-of-Thought to be tagged]:**

First, I analyzed the historical data provided by the user. This data shows descriptions of several movies and their corresponding tags.

I found that the movies tagged as "classic" (like *Tron*, *Jaws*, *The Blues Brothers*, etc.) are all well-known and beloved films released in the 20th century. Movies tagged as "true story" (like *Apollo 13*) are based on real events. This reveals a pattern: the "classic" tag points to older films with a significant era-defining and broad cultural impact.

Next, I analyzed the new movie description to be tagged: "In the years before the Second World War...". This description points to the famous movie *The Sound of Music*.

*The Sound of Music*, released in 1965, tells a household story and achieved immense success and cultural influence worldwide, perfectly matching the characteristics of a "classic" movie that I summarized from the historical data. It is not a sci-fi film, and while it might be partially based on real experiences, its core identity is its classic status.

Therefore, among all candidate tags, "classic" is the most appropriate.

**[Your Output]:**

```
<examine_examples>
First, I analyzed the historical data provided by the user. This
    data shows descriptions of several movies and their
    corresponding tags.
</examine_examples>
<identify_patterns>
I found that the movies tagged as "classic" (like *Tron*, *Jaws*, *
    The Blues Brothers*, etc.) are all well-known and beloved films
    released in the 20th century. Movies tagged as "true story" (
    like *Apollo 13*) are based on real events. This reveals a
    pattern: the "classic" tag points to older films with a
    significant era-defining and broad cultural impact.
```

<analyze_input> <examine_examples> <identify_patterns>
<evaluate_reference> <compare_entities> <synthesize_findings>
<make_decision> <verify_conclusion> <formulate_conclusion>

Figure 9: The refined set of nine primary tags used for annotating reasoning chains. These tags represent the most salient reasoning patterns identified through our clustering analysis.

```
</identify_patterns>
<analyze_input>
Next, I analyzed the new movie description to be tagged: "In the
    years before the Second World War...". This description points
    to the famous movie *The Sound of Music*.
</analyze_input>
<compare_entities>
*The Sound of Music*, released in 1965, tells a household story and
     achieved immense success and cultural influence worldwide,
    perfectly matching the characteristics of a "classic" movie that
     I summarized from the historical data. It is not a sci-fi film,
     and while it might be partially based on real experiences, its
    core identity is its classic status.
</compare_entities>
<make_decision>
Therefore, among all candidate tags, "classic" is the most
    appropriate.
</make_decision>
```

**Now, according to the rules above, please add tags to the [Question] and [Chain-of-Thought] provided below:**

**[Question]:**
{question}

**[Chain-of-Thought to be tagged]:**
{chain_of_thought}

## F.2    REFINED PRIMARY TAGS SET

The final set of primary tags derived from our clustering procedure is listed in Figure 9. These tags were used to annotate the reasoning chains in our dataset.

## G    IMPLEMENTATION GUIDE

To facilitate reproduction, we provide a streamlined guide covering data construction, reward model training, and the multi-stage training.

Data Construction Pipeline. The pipeline follows three steps: (1) Generation & Filtering: Generate $N = 16$ candidate reasoning chains per query using a strong reasoning model. Apply an *Accuracy Filter* to discard incorrect responses, followed by an *LLM Filter* to retain only high-quality logic. (2) Tagging: Perform *Exploratory Tagging* on a subset to cluster patterns, then apply *Restricted Tagging* to the full dataset using the refined tag set (Figure 9).

Personalization Reward Model (PRMU). The PRMU must be trained prior to the RL phase on the preference datasets. (1) Architecture: We initialize the backbone with Skywork-Reward-V2. A key modification is the learnable user embedding matrix $E_u$, corresponding to unique user IDs. (2) Optimization: Both model parameters $\theta_r$ and embeddings $E_u$ are jointly optimized using the Bradley-Terry loss. We recommend validating on held-out preference pairs to ensure the model distinguishes personalized nuances before deployment.

Multi-stage Training. We mainly utilize GSPO for its stability. The training progresses as follows: (1) Stage I: Foundational SFT. Bootstrap the "grammar" of tagged reasoning. Train for 2 epochs with a learning rate of 1e-5. (2) Stage II: Guided RL. Align personalization reasoning. Reward Composition: We set weights $\alpha = \beta = 0.8$ (for structural/factual rewards) and $\gamma = 0.2$ (for PRMU). Use a learning rate of 1e-6. Set the low and high clip ratios for the GSPO to 0.0003 and 0.0004, respectively. Ensure the repetition penalty is tuned to avoid loops. (3) Stage III: Exploratory RL. Refine performance metrics. Initialize with the Stage II checkpoint. Remove $R_{\text{tag}}$ and $R_{\text{PRMU}}$ constraints. Train for 2 epochs to maximize core task performance.

## H    NEW CONSTRUCTED PERSONALIZATION BENCHMARK

To evaluate zero-shot, cross-lingual generalization, we built a benchmark from Dianping, a prominent Chinese user-generated content platform. This appendix details its construction.

### H.1    DATA AND USER PROFILE CREATION

We collected public posts from Dianping and applied rigorous filtering to retain high-quality content, removing short posts, duplicates, and advertisements. From this cleaned dataset, we selected 1,000 users with extensive post histories.

For each user, a profile representing their personal writing style was constructed from their 8 most recent posts (title and content). The 9th most recent post was held out as the ground truth for our evaluation tasks, ensuring a strict zero-shot setting where the test data is unseen.

### H.2    TASK FORMULATION

The benchmark consists of three distinct tasks, with one instance per user for each task, totaling 3,000 evaluation instances. All tasks are conditioned on the user's 8-post profile. As in the LaMP dataset, we use the ROUGE-1 and ROUGE-L metrics for evaluation.

**Dianping-Content (Title $\rightarrow$ Content):** Given the title of the held-out post, the model must generate the full post content in the user's specific style.

**Dianping-Title (Content $\rightarrow$ Title):** The inverse task, where the model generates a stylistically appropriate title from the held-out post's content.

**Dianping-Paraph (Generic $\rightarrow$ Stylized Post):** This task measures stylistic transfer. For each user's held-out post, we first used a general-purpose LLM (GPT-4o) to generate a neutral, generic version based on the original content. The model's task is to rewrite this generic text to match the user's unique style, with the user's original post as the target.

### H.3    BENCHMARK STATISTICS

Key statistics of the final benchmark are summarized in Table 8.

Table 8: Data statistics of the new constructed personalization benchmark.

| Task | Task Type | #Test | #Classes |
|---|---|---|---|
| Dianping-Content | Text generation | 1000 | - |
| Dianping-Title | Text generation | 1000 | - |
| Dianping-Paraph | Text generation | 1000 | - |

