# OpenReview forum: "Tagging the Thought: Unlocking Personalization Reasoning via Reinforcement Learning"
_ICLR.cc/2026/Conference — ICLR 2026 Conference Withdrawn Submission_

### Official Review · Reviewer_yWps · 2025-10-21

**Soundness:** 3
**Presentation:** 3
**Contribution:** 2
**Rating:** 4
**Confidence:** 4

**Summary:**

This paper proposes a personalized reasoning training framework called TagPR to address the limitations of existing LLMs in personalized reasoning. Its core idea is to enable the model to generate explicit and interpretable reasoning steps through a "tagging the thought" approach. This approach involves automatically constructing a semantically labeled reasoning chain dataset, a collaborative training strategy, and a multi-stage reinforcement learning process. Finally, the effectiveness of the proposed method is demonstrated on the LaMP dataset.

**Strengths:**

1. Unlike implicit reasoning, LLM provides structured and interpretable reasoning, which is valuable.
2. The paper is well-written, clearly describing the complete reasoning chain construction and training process.
3. Extensive experiments are conducted, and the results show that the proposed strategy appears to be very effective. Comprehensive ablation experiments are also provided.

**Weaknesses:**

1. Equation 5 is strange; the first term is a composite, while the rest are separate. This requires further discussion.
2. The effectiveness of the proposed strategy is only verified on the LaMP dataset (constructing the reasoning chain data and training the model), and its effectiveness on other datasets is unknown (this does not refer to zero-shot generalization).
3. The rationality of the inference process is unclear. Specifically, it is unclear how the distribution of inference labels (Figure 5, right) indicates the rationality of the cognitive process, nor how Figure 6 demonstrates the correctness of the inference model. Furthermore, more case studies are recommended.
4. The proposed training framework has a large number of parameters, and their selection is unclear. Furthermore, sensitivity analysis is lacking.
5. Mathematical notation needs to be standardized. For example, $\theta$ is used in various models, and loss is represented in two ways: $\mathcal{L}$ and $\mathcal{J}$.

**Questions:**

1. The reward function requires further discussion.
2. Does the proposed framework still work on other datasets?
3. How can we prove the rationality of the reasoning chain, that is, the reasoning that is faithful to the model?
4. How are the numerous hyperparameters selected?

---

> ### Author Response · Authors · 2025-11-21
> **Response to Weakness 1 and Question 1**
>
> Thank you very much for your valuable suggestions! We will try to tackle the concerns one by one.
>
> ---
> **Weakness 1 & Question 1:**
>
> *Equation 5 is strange; the first term is a composite, while the rest are separate. This requires further discussion. & The reward function requires further discussion.*
>
> **Response:**
>
> Thank you for the insightful comment regarding the hybrid structure of Equation 5. We acknowledge that the combination of multiplicative and additive terms requires further clarification, which we have addressed below:
>
> 1.  **Hierarchical Gating Mechanism:** The multiplicative term, $\alpha \cdot (R_v + R_{rep}) \cdot R_f$, acts as a "hard constraint" or gating mechanism. Since $R_f$ is a binary signal (0 or 1) indicating format correctness, multiplying it with content rewards ($R_v$ and $R_{rep}$) ensures that the model is only rewarded for content quality depends on a correct output format. This design enforces a "structure-first" principle, preventing the model from optimizing for content while failing basic structural requirements. Since the value reward $R_v$ serves as the dominant signal for response utility, we explicitly couple it with the format constraint $R_f$.
> 2.  **Training Dynamics:** Empirically, we observe that $R_f$ converges to 1 rapidly during the early stages of training. Consequently, for the majority of the training process, the equation effectively simplifies to a standard additive form: $R \approx \alpha(R_v + R_{rep}) + \beta R_{tag} + \gamma R_{PRMU}$. The multiplicative design is primarily critical for the initial "cold start" phase to guide the model out of format failures.
> 3.  **Ablation Study:** To validate this design choice, we added a new ablation. We changed the hybrid structure reward or removed the format reward, the results are shown below. The results show that our choice is better.
>
> |Method|LaMP-1 ACC ↑| LaMP-1 F1 ↑| LaMP-2 ACC ↑| LaMP-2 F1 ↑|LaMP-3 MAE ↓| LaMP-3 RMSE ↓| LaMP-4 R-1 ↑| LaMP-4 R-L ↑| LaMP-5 R-1 ↑| LaMP-5 R-L ↑| LaMP-7 R-1 ↑| LaMP-7 R-L ↑|
> |---|---|---|---|---|---|---|---|---|---|---|---|---|
> |$R = \alpha \cdot (R_{v} + R_{\text{rep}}) + \beta \cdot R_{\text{tag}} + \gamma \cdot R_{\text{PRMU}}$|0.779|0.779|0.558|0.528|0.239|0.293|0.226|0.208|0.536|0.465|0.553|0.498|
> |$R = (\alpha \cdot R_{v} + \alpha \cdot R_{\text{rep}} + \beta \cdot R_{\text{tag}} + \gamma \cdot R_{\text{PRMU}}) \cdot R_{f}$|0.801|0.801|0.591|0.550|0.223|0.269|0.230|0.211|**0.543**|**0.471**|0.563|0.504|
> |$R = \alpha \cdot (R_{v} + R_{\text{rep}}) \cdot R_{f} + \beta \cdot R_{\text{tag}} + \gamma \cdot R_{\text{PRMU}}$ (Our)|**0.803**|**0.803**|**0.598**|**0.557**|**0.218**|**0.263**|**0.234**|**0.213**|0.542|**0.471**|**0.565**|**0.507**|

---

> ### Author Response · Authors · 2025-11-21
> **Response to Weakness 2 and Question 2**
>
> **Weakness 2 & Question 2:**
>
> *The effectiveness of the proposed strategy is only verified on the LaMP dataset (constructing the reasoning chain data and training the model), and its effectiveness on other datasets is unknown (this does not refer to zero-shot generalization). & Does the proposed framework still work on other datasets?*
>
> **Response:**
>
> Thank you for raising this important question regarding the generalizability of our proposed framework. We acknowledge that verifying effectiveness across diverse scenarios is crucial. In response, we would like to clarify the universality of our design and present additional experimental evidence.
>
> 1. Dataset-Agnostic Framework Design
> Crucially, the TagPR framework is designed to be general and portable. Its core components do not rely on LaMP-specific attributes:
> Universal Data Construction: Our pipeline for constructing reasoning chains is a generalizable "recipe". This process applies to any dataset containing user profiles and interaction histories.
> General Training Strategy: The "SFT + Two-Stage RL" strategy optimizes for universal qualities—factuality, format, logic, and user consistency—rather than dataset-specific heuristics.
> Transferable Reasoning Patterns: The "reasoning tags" derived by our method represent high-level personalization patterns that are applicable across different tasks. This allows our SFT model and tags to be adapted to new personalization tasks directly via RL, significantly reducing the need to reconstruct reasoning chains from scratch.
> 2. Additional Verification on LongLaMP[1]:
> To empirically address your concern, we have extended our experiments to the LongLaMP dataset. Building upon the SFT-trained model, we conduct retraining on the LongLaMP dataset using our proposed RL approach. LongLaMP presents a different challenge (handling extensive user histories) compared to standard LaMP. Our method successfully adapted to this new dataset and achieved state-of-the-art performance. These results shown below confirm that our framework remains effective on other datasets.
>
> |Method|TopicWriting R-1 ↑|TopicWriting R-L ↑|ProductReview R-1 ↑|ProductReview R-L ↑|AbstractGeneration R-1 ↑|AbstractGeneration R-L ↑|
> |---|---|---|---|---|---|---|
> |GPT-4o|0.294| 0.140  |0.330|0.157 |0.372|0.200 |
> |Gemini-2.5-Pro|0.270| 0.132  |0.372|0.171 |0.398|0.215 |
> |Qwen3-8B-thinking (Base)|0.271|0.124|0.321|0.150|0.351 |0.182 |
> |TagPR (Zero-shot Generalization)|0.303|0.154|0.377|0.175|0.404|0.221|
> |TagPR (Trained on LongLaMP)|**0.376**|**0.173**|**0.434**|**0.197**|**0.478**|**0.256**|
>
> [1] LongLaMP: A Benchmark for Personalized Long-form Text Generation, Arxiv, 2024.

---

> ### Author Response · Authors · 2025-11-21
> **Response to Weakness 3 and Question 3**
>
> **Weakness 3 & Question 3:**
>
> *The rationality of the inference process is unclear. Specifically, it is unclear how the distribution of inference labels (Figure 5, right) indicates the rationality of the cognitive process, nor how Figure 6 demonstrates the correctness of the inference model. Furthermore, more case studies are recommended. & How can we prove the rationality of the reasoning chain, that is, the reasoning that is faithful to the model?*
>
> **Response:**
>
> We genuinely appreciate your insightful questions regarding the interpretability and validity of the reasoning process. We acknowledge that verifying the "faithfulness" of the reasoning chain is a core challenge in CoT research. Below, we clarify the implications of Figures 5 and 6, and outline our multi-dimensional approach to validating the reasoning chain. As suggested, we have expanded the Appendix with additional case studies.
>
> 1. Rationality of the Cognitive Process (Figure 5):
> The label distribution in Figure 5 is not random; rather, it serves as evidence that the model has internalized a structured, hierarchical cognitive workflow. The dominance of \<analyze_input\> (38.2%) and \<examine_examples\> (27.7%) indicates that the model prioritizes information gathering and context understanding. This mirrors a rational human problem-solving approach where observation must precede deduction. The distribution reflects a coherent transition from concrete analysis to abstract synthesis (\<identify_patterns\>, 12.4%) and final execution (\<make_decision\>, 17.1%). This demonstrates that the framework successfully guides the model through a systematic logic flow rather than relying on intuitive shortcuts. We will clarify this logical hierarchy in the final paper.
> 2. Indication of Reasoning Chain Correctness (Figure 6):
> Figure 6 provides qualitative evidence of a fundamental behavioral shift that drives model correctness.
> From Description to Execution: The word clouds visualize a transition from the baseline’s uncertain, descriptive language (e.g., "think," "maybe") to TagPR’s action-oriented execution (e.g., "examine," "identify"). This indirectly indicates the effectiveness of personalized reasoning
> 3. Proving the Rationality and Faithfulness of the Reasoning Chain:
>
>   - Outcome Consistency (Quantitative Proof): It would be unlikely for the model to achieve SOTA results across all six LaMP tasks (Table 1 in Section 4.2) or to generalize effectively to the unseen Dianping dataset (Table 2 in Section 4.4). We believe the consistency of these results provides evidence supporting the validity of the Reasoning Chain.
>
>   - Qualitative Verification (Case Studies): Thanks for your insightful suggestion, we have expanded the Appendix with additional case studies from diverse tasks. These cases explicitly trace how the model derives conclusions from specific examples where baselines fail. Here we present a compressed comparative reasoning chain case.
>
> > **Qwen3-8B-thinking**: \<think\>… Let's break down the description. The key elements here are a bankrupt USA, a telethon to prevent repossession by Native Americans... The mention of a bankrupt USA and repossession by Native Americans suggests a critique of economic systems or social issues. That leans towards social commentary. ... This one is about a specific economic crisis... It might not be a dystopia but more of a satirical or social commentary. ... So, the most fitting tag here is "social commentary". \</think\> social commentary
>
> > **TagPR**: \<think\>...\<examine_examples\> First, let‘s look at the examples: ... \</examine_examples\> \<identify_patterns\> From these examples, we see that the tag “dystopia” is used for movies that are set in a futuristic, oppressive, or bleak society. \</identify_patterns\> ... \<compare_entities\> This description fits the pattern of the other dystopian movies listed. The key elements are: … \</compare_entities\>... \<make_conclusion\> Given the examples and the description, "dystopia" is the most appropriate tag. \</make_conclusion\> \</think\> dystopia

---

> ### Author Response · Authors · 2025-11-21
> **Response to Weakness 4, Question 4 and Weakness 5**
>
> **Weakness 4 & Question 4:**
>
> *The proposed training framework has a large number of parameters, and their selection is unclear. Furthermore, sensitivity analysis is lacking. & How are the numerous hyperparameters selected?*
>
> **Response:**
>
> We sincerely thank you for this constructive comment. We clarified the parameter selection process more clearly and added the sensitivity analysis.
>
> 1. Parameter Selection: For standard training parameters in the SFT and RL stages, we aligned our configurations with established practices in recent literature [2, 3, 4]. The batch size was optimized based on computational efficiency and hardware memory constraints.
>
> 2. Reward Weight Rationale: The reward coefficients ($\alpha=0.8, \beta=0.8, \gamma=0.2$) were determined through preliminary empirical tuning. Our rationale is to prioritize structural integrity and logical correctness over personalization. Assigning higher weights to reasoning and format ensures the model establishes correct reasoning chains first, while a lower weight on personalization prevents the model from hallucinating or sacrificing factuality to merely align with user preferences and reward hacking.
>
> 3. Sensitivity Analysis: Following your valuable suggestion, we have conducted a hyperparameter sensitivity analysis on the reward weights. The results are shown below. The model performance remains robust within a reasonable range of parameter variations and our current configuration yields the most balanced performance.
>
> |Method|LaMP-1 ACC ↑| LaMP-1 F1 ↑| LaMP-2 ACC ↑| LaMP-2 F1 ↑|LaMP-3 MAE ↓| LaMP-3 RMSE ↓| LaMP-4 R-1 ↑| LaMP-4 R-L ↑| LaMP-5 R-1 ↑| LaMP-5 R-L ↑| LaMP-7 R-1 ↑| LaMP-7 R-L ↑|
> |---|---|---|---|---|---|---|---|---|---|---|---|---|
> |$\alpha=1.0, \beta=1.0, \gamma=1.0$|0.797|0.797|0.580|0.552|0.226|0.271|0.226|0.208|0.540|0.469|0.563|0.504|
> |$\alpha=0.2, \beta=0.2, \gamma=0.8$|0.789|0.788|0.572|0.531|0.230|0.272|0.225|0.207|0.532|0.459|0.561|0.500|
> |$\alpha=0.8, \beta=0.2, \gamma=0.2$|0.789|0.789|0.575|0.536|0.229|0.272|0.227|0.208|0.539|0.466|0.564|0.505|
> |$\alpha=0.2, \beta=0.8, \gamma=0.2$|0.785|0.785|0.573|0.535|0.235|0.275|0.225|0.206|0.534|0.459|0.560|0.498|
> |$\alpha=0.8, \beta=0.8, \gamma=0.4$|0.801|0.800|0.594|0.553|**0.216**|**0.262**|0.231|0.212|**0.542**|0.470|0.564|0.505|
> |$\alpha=0.8, \beta=0.8, \gamma=0.2$ (Our)|**0.803**|**0.803**|**0.598**|**0.557**|0.218|0.263|**0.234**|**0.213**|**0.542**|**0.471**|**0.565**|**0.507**|
>
> [2] Deepseek-r1 incentivizes reasoning in llms through reinforcement learning, Nature, 2025.
>
> [3] Qwen3 Technical Report, Arxiv, 2025.
>
> [4] Group Sequence Policy Optimization, Arxiv, 2025.
>
> ---
> **Weakness 5**
>
> *Mathematical notation needs to be standardized. For example, $\theta$ is used in various models, and loss is represented in two ways:  $\mathcal{J}$ and $\mathcal{L}$.*
>
> **Response:**
>
> We sincerely thank you for pointing out the inconsistency in our mathematical notation. We have carefully revised the manuscript to standardize the notation throughout. Specifically, we have unified the representation of the loss function to $\mathcal{J}$ and ensured that $\theta$ is used consistently across different models. We have also conducted a thorough proofreading to correct any remaining ambiguous notations.
>
> ---
> Thank you again for the thorough and constructive feedback! We hope our responses have adequately addressed your concerns. We would be deeply grateful if you could consider raising your score based on these clarifications. We look forward to further discussion if additional questions arise.

---

> > ### Comment · Reviewer_yWps · 2025-11-25
> >
> > Thank you for the authors' detailed response; it resolved many of my questions. Regarding the sensitivity analysis, I appreciate the demonstration of performance under different hyperparameter settings. However, I would like to see the changes in model performance under control variables, as this would provide a better understanding of the effects of various rewards. Furthermore, I hope the authors can provide a more in-depth explanation of the observed trade-offs and discuss their rationale.

---

> ### Author Response · Authors · 2025-11-27
> **Respone to Control Variable Analysis (Part 1/2)**
>
> We strictly appreciate your constructive feedback and are glad to hear that our previous responses have resolved many of your concerns!
>
> ---
>
> We greatly appreciate your suggestion to conduct a control variable analysis. We agree that observing the performance changes under controlled variables provides a deeper insight into the specific contribution of each reward component.
>
> As suggested, we performed a comprehensive sensitivity analysis by combining results from the original ablation studies, prior supplements, and new experiments. We varied one hyperparameter at a time ($\alpha$ for Verifiable Reward, $\beta$ for Tag Reward, and $\gamma$ for Personalization Reward) while keeping the others fixed at their optimal values. The detailed results on the LaMP benchmark are presented below, followed by our discussion on the rationale and trade-offs.
>
> **Controlled Variable Experimental Results:**
>
> (A) Sensitivity to Verifiable Reward ($\alpha$) while keeping $\beta=0.8$ and $\gamma=0.2$:
>
> |Method|LaMP-1 ACC ↑| LaMP-1 F1 ↑| LaMP-2 ACC ↑| LaMP-2 F1 ↑|LaMP-3 MAE ↓| LaMP-3 RMSE ↓| LaMP-4 R-1 ↑| LaMP-4 R-L ↑| LaMP-5 R-1 ↑| LaMP-5 R-L ↑| LaMP-7 R-1 ↑| LaMP-7 R-L ↑|
> |---|---|---|---|---|---|---|---|---|---|---|---|---|
> |$\alpha=0.0, \beta=0.8, \gamma=0.2$|0.701|0.701|0.457|0.417|0.345|0.892|0.163|0.149|0.431|0.371|0.433|0.385|
> |$\alpha=0.2, \beta=0.8, \gamma=0.2$|0.785|0.785|0.573|0.535|0.235|0.275|0.225|0.206|0.534|0.459|0.560|0.498|
> |$\alpha=0.4, \beta=0.8, \gamma=0.2$|0.788|0.788|0.583|0.542|0.228|0.270|0.229|0.208|0.535|0.461|0.561|0.501|
> |$\alpha=0.6, \beta=0.8, \gamma=0.2$|0.797|0.798|0.596|0.555|0.216|0.262|0.231|0.211|0.538|0.468|0.566|0.507|
> |$\alpha=0.8, \beta=0.8, \gamma=0.2$|0.803|0.803|0.598|0.557|0.218|0.263|0.234|0.213|0.542|0.471|0.565|0.507|
>
> (B) Sensitivity to Tag Reward ($\beta$) while keeping $\alpha=0.8$ and $\gamma=0.2$:
>
> |Method|LaMP-1 ACC ↑| LaMP-1 F1 ↑| LaMP-2 ACC ↑| LaMP-2 F1 ↑|LaMP-3 MAE ↓| LaMP-3 RMSE ↓| LaMP-4 R-1 ↑| LaMP-4 R-L ↑| LaMP-5 R-1 ↑| LaMP-5 R-L ↑| LaMP-7 R-1 ↑| LaMP-7 R-L ↑|
> |---|---|---|---|---|---|---|---|---|---|---|---|---|
> |$\alpha=0.8, \beta=0.0, \gamma=0.2$|0.749|0.749|0.545|0.511|0.272|0.595|0.197|0.183|0.506|0.441|0.524|0.469|
> |$\alpha=0.8, \beta=0.2, \gamma=0.2$|0.789|0.789|0.575|0.536|0.229|0.272|0.227|0.208|0.539|0.466|0.564|0.505|
> |$\alpha=0.8, \beta=0.4, \gamma=0.2$|0.799|0.799|0.589|0.548|0.220|0.263|0.231|0.210|0.539|0.469|0.563|0.505|
> |$\alpha=0.8, \beta=0.6, \gamma=0.2$|0.801|0.801|0.595|0.553|0.216|0.262|0.235|0.212|0.542|0.470|0.564|0.505|
> |$\alpha=0.8, \beta=0.8, \gamma=0.2$|0.803|0.803|0.598|0.557|0.218|0.263|0.234|0.213|0.542|0.471|0.565|0.507|
>
> (C) Sensitivity to Personalization Reward ($\gamma$) while keeping $\alpha=0.8$ and $\beta=0.8$:
>
> |Method|LaMP-1 ACC ↑| LaMP-1 F1 ↑| LaMP-2 ACC ↑| LaMP-2 F1 ↑|LaMP-3 MAE ↓| LaMP-3 RMSE ↓| LaMP-4 R-1 ↑| LaMP-4 R-L ↑| LaMP-5 R-1 ↑| LaMP-5 R-L ↑| LaMP-7 R-1 ↑| LaMP-7 R-L ↑|
> |---|---|---|---|---|---|---|---|---|---|---|---|---|
> |$\alpha=0.8, \beta=0.8, \gamma=0.0$|0.768|0.769|0.557|0.514|0.246|0.393|0.205|0.190|0.522|0.453|0.545|0.490|
> |$\alpha=0.8, \beta=0.8, \gamma=0.2$|0.803|0.803|0.598|0.557|0.218|0.263|0.234|0.213|0.542|0.471|0.565|0.507|
> |$\alpha=0.8, \beta=0.8, \gamma=0.4$|0.801|0.800|0.594|0.553|0.216|0.262|0.231|0.212|0.542|0.470|0.564|0.505|
> |$\alpha=0.8, \beta=0.8, \gamma=0.6$|0.781|0.782|0.569|0.535|0.241|0.301|0.221|0.205|0.539|0.466|0.552|0.496|
> |$\alpha=0.8, \beta=0.8, \gamma=0.8$|0.778|0.778|0.563|0.525|0.237|0.288|0.219|0.204|0.536|0.464|0.550|0.496|

---

> ### Author Response · Authors · 2025-11-27
> **Respone to Control Variable Analysis (Part 2/2)**
>
> **Discussion on Rationale and Trade-offs:**
>
> Based on the controlled variable analysis presented above, we derive three critical observations regarding the interaction dynamics of our reward components. These findings validate our initial design logic and the necessity of the chosen hyperparameter configuration ($\alpha=0.8, \beta=0.8, \gamma=0.2$):
>
> 1. The Primacy of Verifiable Reward ($\alpha$):
> As evidenced in Table (A), the model is most sensitive to the Verifiable Reward. The severe performance degradation at $\alpha=0.0$ confirms that factual correctness serves as the cornerstone of our RL framework. The performance gain stabilizes around $\alpha=0.8$, suggesting that once a sufficient baseline of correctness is established, further improvements are driven by structural and personalization signals rather than factual penalties alone.
>
> 2. The Efficacy of Tag Reward ($\beta$):
> Table (B) identifies the Tag Reward as a vital structural scaffold. While the model is less sensitive to $\beta$ than to $\alpha$, the absence of tag constraints leads to suboptimal reasoning patterns. The performance plateau observed as $\beta$ increases indicates that the signal is robust. It successfully enforces the necessary personalization reasoning schema (the "Tagging" process), effectively instituting process-level supervision by transforming implicit inference into a structured, interpretable sequence. The results indicate that this explicit constraint is essential for preventing the model from reverting to generic generation patterns, thereby ensuring that the reasoning remains grounded in the user's specific context before the final response is formulated.
>
> 3. The Trade-off in Personalization ($\gamma$) and Reward Hacking:
> Table (C) reveals a distinct "inverted-U" trajectory for the Personalization Reward, contrasting with the monotonic trends of $\alpha$ and $\beta$. While $\gamma=0.2$ effectively aligns the model with user preferences, excessive weights ($\gamma \ge 0.6$) lead to performance degradation. This aligns with the risk of reward hacking: when the personalization signal becomes too dominant, the model begins to prioritize mimicking historical user style or patterns at the expense of factual accuracy and input fidelity.
>
> **Conclusion:**
>
> Our analysis confirms that Verifiable Reward ($\alpha$) and Tag Reward ($\beta$) must act as the primary drivers to ensure valid reasoning. Personalization Reward ($\gamma$) functions best as a regularizer that guides the solution space toward user preference without overriding the fundamental logic. The selected configuration ($\alpha=0.8, \beta=0.8, \gamma=0.2$) represents the optimal balance, preventing "over-personalization" while maximizing the synergy between structured reasoning and user alignment.
>
> ---
>
> We sincerely appreciate your suggestion to conduct this control variable analysis, as it has significantly strengthened the empirical validity of our work. We hope these new results fully resolve your concerns, and we would be grateful if you could consider raising your score accordingly. We remain available to address any further questions or concerns if you have any!

---

> > ### Comment · Reviewer_yWps · 2025-11-27
> >
> > Thank you for your prompt response. I have no further questions. I suggest incorporating these discussion findings into the final manuscript. Accordingly, I have increased the rating. Good luck!

---

> > > ### Author Response · Authors · 2025-11-27
> > >
> > > We sincerely thank you for your time and support! We will incorporate these findings into the final manuscript. Your feedback has been incredibly helpful in improving our work！

---

### Official Review · Reviewer_mT6b · 2025-10-30

**Soundness:** 3
**Presentation:** 2
**Contribution:** 2
**Rating:** 4
**Confidence:** 2

**Summary:**

This paper introduces TagPR, a novel framework that enhances LLMs ability for personalization reasoning. TagPR adopts a “tagging the thought” approach that first builds a structured dataset by automatically labeling reasoning chains, then trains the model through Supervised Fine-Tuning and multi-stage Reinforcement Learning. The reinforcement process is guided by a composite reward combining tag-based constraints and a Personalization Reward Model with User Embeddings (PRMU) to align reasoning with user-specific logic. Experiments on public and self-constructed datasets show that TagPR achieves state-of-the-art performance.

**Strengths:**

1. The authors have developed a new dataset designed to advance future studies on LLM reasoning, providing valuable resources for the research community.
2. The two-stage approach with SFT + RL is technically sound.

**Weaknesses:**

1. I am not certain about the motivation of the paper which uses tagging to address personalization. Why would this be an effective approach / direction?
2. Evaluation protocols are not clearly mentioned. For example, how is the accuracy and F1 calculated.

**Questions:**

1. On what basis were the reward weights (α, β, γ) chosen in Eq (5), and how sensitive is the model’s performance to these values?
2. How does the model maintain personalization fidelity and reasoning consistency when the personalization and tag rewards are removed during exploratory RL?
3. I am not very familiar with the benchmark and got quite confused about how the evaluation is done once the LLM is trained. Is it using logprob or generations from LLMs?

---

> ### Author Response · Authors · 2025-11-21
> **Response to Weakness 1 and Weakness 2**
>
> We sincerely thank your insightful issues raised regarding our work. Below, we provide point-by-point responses to address your concerns and offer further clarification.
>
> ---
> **Weakness 1:**
>
> *I am not certain about the motivation of the paper which uses tagging to address personalization. Why would this be an effective approach / direction?*
>
> **Response:**
>
> Thank you for this insightful question. We acknowledge that the original manuscript could have stated the technical motivation behind "tagging" more clearly.
>
> Our motivation stems from the limitation of outcome-based rewards, which are often sparse and indirect for complex personalization tasks. A correct final answer does not necessarily imply a correct reasoning process. To address this, we use tagging to transform implicit reasoning steps into an explicit structure, enabling process-level supervision.
>
> Specifically, tagging offers two key advantages:
>
> **Guidance via Process Rewards:** By explicitly defining steps like \<examine_examples\> and \<identify_patterns\>, we can apply "Tag Reward" (as detailed in Section 3.4) to verify intermediate logic. This prevents the model from learning spurious correlations and ensures the reasoning path is as valid as the final result.
>
> **Interpretability:** Tags force the model to externalize its thought process, naturally generating a self-explanatory chain of thought. This transparency is crucial for building trust in personalized systems.
>
> We will revise the Introduction and Methodology sections to explicitly highlight these motivations.
>
> ---
> **Weakness 2:**
>
> *Evaluation protocols are not clearly mentioned. For example, how is the accuracy and F1 calculated.*
>
> **Response:**
>
> We sincerely thank you for highlighting the need for greater clarity regarding our evaluation protocols. We recognize the importance of explicitly defining these metrics to ensure reproducibility. In our experiments, we strictly followed the official protocols of the LaMP Benchmark.
>
> While we initially referenced these settings in Appendix, we will revise to provide a more detailed breakdown of the task definitions and their corresponding calculation methods:
>
> 1. Classification Tasks (LaMP-1 & LaMP-2)
> Task Descriptions:
> **LaMP-1 (Personalized Citation):** A binary classification task where the model determines which of two candidate references an author is likely to cite based on their profile.
> **LaMP-2 (Personalized Movie Tagging):** A multi-class classification task requiring the model to select the most appropriate tag from 15 candidates for a movie, conditioned on the user’s tagging history.
> Metrics & Calculation: We report Accuracy and F1-Score.
> For **Accuracy**, we calculate the ratio of correctly predicted samples to the total samples.
> For **F1-Score**: In LaMP-1 (binary), we report the standard F1 for the positive class. In LaMP-2 (multi-class), we follow the benchmark standard and report Macro-F1. This is calculated as the arithmetic mean of the per-class F1 scores, ensuring that the model’s performance on all categories is evaluated equally.
>
> 2. Rating Task (LaMP-3)
> Task Description: **LaMP-3 (Personalized Product Rating)** is an ordinal classification task. The model predicts a specific integer rating (1–5 stars) for a product based on the user's historical reviews.
> Metrics: We employ **MAE (Mean Absolute Error)** and **RMSE (Root Mean Square Error)**. These metrics are standard for quantifying the deviation between predicted scores and ground-truth user ratings.
>
> 3. Text Generation Tasks (LaMP-4, 5, & 7)
> Task Descriptions: These tasks involve generating text tailored to a user's specific style:
> **LaMP-4:** Personalized news headline generation.
> **LaMP-5:** Personalized scholarly title generation.
> **LaMP-7:** Personalized tweet paraphrasing.
> Metrics: We utilize ROUGE-1 and ROUGE-L.
> **ROUGE-1:** measures the overlap of unigrams between the generated text and the reference to assess content adequacy.
> **ROUGE-L:** is based on the Longest Common Subsequence, evaluating fluency and structural similarity between the generated text and the reference.

---

> ### Author Response · Authors · 2025-11-21
> **Response to Question 1 and Question 2**
>
> **Question 1:**
>
> *On what basis were the reward weights ($\alpha, \beta, \gamma$) chosen in Eq (5), and how sensitive is the model’s performance to these values?*
>
> **Response:**
>
> Thank you for this thoughtful question. Our current weight assignment ($\alpha=0.8, \beta=0.8, \gamma=0.2$) follows a specific design rationale: prioritizing structural integrity over personalization. High weights on reasoning and format ensure the model first establishes logically correct reasoning chains, while a lower weight on personalization prevents the model from sacrificing factuality to over-align with user preferences and reward hacking.
>
> Following your advice, we have included a hyperparameter sensitivity analysis about the reward weights. The results are shown below. The model performance remains robust within a reasonable range of parameter variations and our current configuration yields the most balanced performance.
>
>
> |Method|LaMP-1 ACC ↑| LaMP-1 F1 ↑| LaMP-2 ACC ↑| LaMP-2 F1 ↑|LaMP-3 MAE ↓| LaMP-3 RMSE ↓| LaMP-4 R-1 ↑| LaMP-4 R-L ↑| LaMP-5 R-1 ↑| LaMP-5 R-L ↑| LaMP-7 R-1 ↑| LaMP-7 R-L ↑|
> |---|---|---|---|---|---|---|---|---|---|---|---|---|
> |$\alpha=1.0, \beta=1.0, \gamma=1.0$|0.797|0.797|0.580|0.552|0.226|0.271|0.226|0.208|0.540|0.469|0.563|0.504|
> |$\alpha=0.2, \beta=0.2, \gamma=0.8$|0.789|0.788|0.572|0.531|0.230|0.272|0.225|0.207|0.532|0.459|0.561|0.500|
> |$\alpha=0.8, \beta=0.2, \gamma=0.2$|0.789|0.789|0.575|0.536|0.229|0.272|0.227|0.208|0.539|0.466|0.564|0.505|
> |$\alpha=0.2, \beta=0.8, \gamma=0.2$|0.785|0.785|0.573|0.535|0.235|0.275|0.225|0.206|0.534|0.459|0.560|0.498|
> |$\alpha=0.8, \beta=0.8, \gamma=0.4$|0.801|0.800|0.594|0.553|**0.216**|**0.262**|0.231|0.212|**0.542**|0.470|0.564|0.505|
> |$\alpha=0.8, \beta=0.8, \gamma=0.2$ (Our)|**0.803**|**0.803**|**0.598**|**0.557**|0.218|0.263|**0.234**|**0.213**|**0.542**|**0.471**|**0.565**|**0.507**|
>
> ---
> **Question 2:**
>
> *How does the model maintain personalization fidelity and reasoning consistency when the personalization and tag rewards are removed during exploratory RL?*
>
> **Response:**
>
> We appreciate this insightful question regarding the stability of the model’s reasoning capabilities during the final training phase. The decision to remove explicit personalization and tag constraints during the Exploratory RL stage is a deliberate design choice aimed at balancing structural adherence with outcome optimization.
>
> We address your concern through the following three perspectives:
>
> - Internalization via Guided RL: The preceding Guided RL stage is extensive (13 epochs) and utilizes comprehensive reward signals (Tag & PRMU) to firmly establish the "grammar" of structured personalization. By the time the model enters the Exploratory phase, the reasoning workflow is no longer just a constraint but has been internalized as the model's intrinsic policy.
>
> - Avoiding Local Optima: Continuous enforcement of rigid process-based rewards can lead to a performance plateau, where the model prioritizes satisfying format constraints over maximizing answer quality. The Exploratory RL stage acts as a refinement phase (trained for only 2 epochs). By shifting the objective from "process correctness" to "outcome superiority," we encourage the model to optimize its policy within the learned structural framework, avoiding the trap of low-entropy, mechanical generation.
>
> - Implicit Consistency: Although explicit tag rewards are removed, the Foundation Reward (correctness/ROUGE) implicitly reinforces the reasoning structure. Since the model has learned that tag-based personalization reasoning is the most effective path to a correct answer, it naturally persists in using this structure to maximize the reward, even without direct enforcement.
>
> Qualitative analysis (Table 1 in Section 4.3) confirms that the model retains its structured reasoning behavior after this phase, while quantitative results demonstrate that this "unshackling" boosts final performance.

---

> ### Author Response · Authors · 2025-11-21
> **Response to Question 3**
>
> **Question 3:**
>
> *I am not very familiar with the benchmark and got quite confused about how the evaluation is done once the LLM is trained. Is it using logprob or generations from LLMs?*
>
> **Response:**
>
> We appreciate your query. To clarify, all our evaluations rely on the model's generations, strictly following the official LaMP Benchmark. We do not evaluate based on log-probabilities.
>
> We will revise the manuscript to more explicitly describe the task definitions and calculation methods:
>
> 1. Classification Tasks (LaMP-1 & LaMP-2)
> Task Descriptions:
> **LaMP-1 (Personalized Citation):** A binary classification task where the model determines which of two candidate references an author is likely to cite based on their profile.
> **LaMP-2 (Personalized Movie Tagging):** A multi-class classification task requiring the model to select the most appropriate tag from 15 candidates for a movie, conditioned on the user’s tagging history.
> Metrics & Calculation: We report Accuracy and F1-Score.
> For **Accuracy**, we calculate the ratio of correctly predicted samples to the total samples.
> For **F1-Score**: In LaMP-1 (binary), we report the standard F1 for the positive class. In LaMP-2 (multi-class), we follow the benchmark standard and report Macro-F1. This is calculated as the arithmetic mean of the per-class F1 scores, ensuring that the model’s performance on all categories (including rare ones) is evaluated equally.
>
> 2. Rating Task (LaMP-3)
> Task Description: **LaMP-3 (Personalized Product Rating)** is an ordinal classification task. The model predicts a specific integer rating (1–5 stars) for a product based on the user's historical reviews.
> Metrics: We employ **MAE (Mean Absolute Error)** and **RMSE (Root Mean Square Error)**. These metrics are standard for quantifying the deviation between predicted scores and ground-truth user ratings.
>
> 3. Text Generation Tasks (LaMP-4, 5, & 7)
> Task Descriptions: These tasks involve generating text tailored to a user's specific style:
> **LaMP-4:** Personalized news headline generation.
> **LaMP-5:** Personalized scholarly title generation.
> **LaMP-7:** Personalized tweet paraphrasing.
> Metrics: We utilize ROUGE-1 and ROUGE-L.
> **ROUGE-1:** measures the overlap of unigrams to assess content adequacy.
> **ROUGE-L:** is based on the Longest Common Subsequence, evaluating fluency and structural similarity between the generated text and the reference.
>
> ---
> We sincerely appreciate your thoughtful comments and time in reviewing our response! We hope our clarifications have adequately addressed your concerns, and we would be grateful if you could consider raising your score accordingly. We remain available to address any further questions or concerns if you have any.

---

### Official Review · Reviewer_mHx1 · 2025-11-01

**Soundness:** 3
**Presentation:** 3
**Contribution:** 2
**Rating:** 4
**Confidence:** 4

**Summary:**

This paper addresses the limitation of large language models (LLMs) in personalization reasoning and proposes TagPR, a framework that uses a data-driven pipeline to create tagged reasoning chains and a synergistic training strategy (SFT + multi-stage RL). It demonstrates TagPR achieves state-of-the-art results on the LaMP benchmark and a self-constructed Dianping benchmark, outperforming base models and even larger proprietary LLMs while showing strong generalization.

**Strengths:**

1. The trained model enhances LLMs’ personalization reasoning, achieving state-of-the-art performance on LaMP benchmark.
2. The proposed training framework (SFT + multi-stage RL) with sophisticated reward design is insightful for relevant tasks.

**Weaknesses:**

1. The framework relies on high-quality initial reasoning chains generated by Qwen3 Thinking model and GPT-4o, inducing huge cost on constructing data pipelines.
2. The two-stage RL training (especially the guided phase with multiple reward components) involves more tuning, increasing implementation and reproduction difficulties.
3. The training data and evaluation data are mainly related to the LaMP benchmark, regardless of the private Dianping benchmark, making the generalization capability of this model on other related benchmark questionable.

**Questions:**

1. Could you elaborate on how the choice of the 9 primary tags in the tagged reasoning chains dataset was determined?
2. Will the method benefit more from more careful choices on the reward composition?
3. How well does the PRMU improve the model performance?

---

> ### Author Response · Authors · 2025-11-21
> **Response to Weakness 1 and Weakness 2**
>
> We sincerely thank you for the insightful comments and valuable questions regarding our work. Below, we provide a point-by-point response to address the concerns raised and further demonstrate the contributions of our approach.
>
> ---
> **Weakness 1:**
>
> *The framework relies on high-quality initial reasoning chains generated by Qwen3 Thinking model and GPT-4o, inducing huge cost on constructing data pipelines.*
>
> **Response:**
>
> We genuinely appreciate your insightful concern regarding the cost of the data pipeline. We acknowledge that the initial construction involves a significant one-time investment. However, the primary goal of this phase is to distill high-quality, generalizable thinking paradigms (Tags set), rather than merely generating data for a single task.
>
> Crucially, this expensive generation process can be not required for downstream adaptation. To validate this, we have conducted supplementary experiments on the LongLaMP dataset [1]. The results demonstrate that our distilled Tags and the SFT-initialized model can be efficiently transferred to new personalized tasks via RL directly, bypassing the need to reconstruct massive reasoning chains from scratch.
>
> |Method|TopicWriting R-1 ↑|TopicWriting R-L ↑|ProductReview R-1 ↑|ProductReview R-L ↑|AbstractGeneration R-1 ↑|AbstractGeneration R-L ↑|
> |---|---|---|---|---|---|---|
> |GPT-4o|0.294| 0.140  |0.330|0.157 |0.372|0.200 |
> |Gemini-2.5-Pro|0.270| 0.132  |0.372|0.171 |0.398|0.215 |
> |Qwen3-8B-thinking (Base)|0.271|0.124|0.321|0.150|0.351 |0.182 |
> |TagPR (Zero-shot Generalization)|0.303|0.154|0.377|0.175|0.404|0.221|
> |TagPR (Trained on LongLaMP)|**0.376**|**0.173**|**0.434**|**0.197**|**0.478**|**0.256**|
>
> To further mitigate these costs for the community, we commit to open-sourcing our curated dataset, the extracted Tags Set, and the SFT-initialized model, thereby providing a reusable foundation for future research.
>
> [1] LongLaMP: A Benchmark for Personalized Long-form Text Generation, Arxiv, 2024.
>
> ---
> **Weakness 2:**
>
> *The two-stage RL training (especially the guided phase with multiple reward components) involves more tuning, increasing implementation and reproduction difficulties.*
>
> **Response:**
>
> We appreciate your concern regarding the implementation complexity and hyperparameter tuning. We would like to clarify that the multi-component reward design is essential to guide the model across multiple dimensions—such as personalization, logical coherence, and factual accuracy, as shown in our Ablation Study (Table 1 in Section 4.3).
>
> To fully address the concern about reproducibility, we include our code in the Supplementary Material and are committed to releasing all source code and training scripts. Furthermore, we will update the Appendix with a more detailed "implementation guide"  to facilitate easy reproduction by the community.

---

> ### Author Response · Authors · 2025-11-21
> **Response to Weakness 3 and Question 1**
>
> **Weakness 3:**
>
> *The training data and evaluation data are mainly related to the LaMP benchmark, regardless of the private Dianping benchmark, making the generalization capability of this model on other related benchmark questionable.*
>
> **Response:**
>
> Thank you for highlighting the importance of generalization. We would like to clarify our evaluation and present additional evidence to address this concern:
>
> - Standard Benchmark (LaMP): We primarily utilized LaMP because it is the recognized gold standard in personalized generation, comprising 7 diverse tasks. This allows for fair comparison with SOTA methods.
>
> - Zero-Shot Generalization (Dianping): We specifically constructed the Dianping dataset to rigorously test generalization in Section 4.4. The model’s strong performance here effectively demonstrates its robustness beyond the training distribution. We also commit to open-sourcing this dataset to facilitate future community research.
>
> - Supplementary Generalization Evaluation (LongLaMP & PGraphRAG [2]): Following your constructive suggestion, we have extended our evaluation to two third-party public benchmarks to further verify our method's effectiveness across different data distributions.
>
> First, we evaluated the model on LongLaMP. The experimental results are consistent with our previous findings, confirming the model's capability in handling long-context personalized tasks.
>
> |Method|TopicWriting R-1 ↑|TopicWriting R-L ↑|ProductReview R-1 ↑|ProductReview R-L ↑|AbstractGeneration R-1 ↑|AbstractGeneration R-L ↑|
> |---|---|---|---|---|---|---|
> |Qwen3-8B-nothinking|0.282|0.134 |0.342|0.154 |0.382 |0.203 |
> |Qwen3-8B-thinking| 0.271|0.124 |0.321|0.150|0.351 |0.182 |
> |Qwen3-32B-nothinking|0.292| 0.134  |0.354|0.159 |0.384 |0.202 |
> |Qwen3-32B-thinking|0.275| 0.119  |0.320|0.144 |0.346|0.177 |
> |GPT-4o|0.294| 0.140  |0.330|0.157 |0.372|0.200 |
> |Gemini-2.5-Pro|0.270| 0.132  |0.372|0.171 |0.398|0.215 |
> |TagPR (Our)|**0.303**|**0.154**|**0.377**|**0.175**|**0.404**|**0.221**|
>
>
> Second, we conducted experiments on the PGraphRAG benchmark using the Amazon Review test subset, retaining users with at least one history review. The dataset is characterized by extremely sparse historical profiles. Despite this data scarcity, TagPR achieves superior performance on Title Generation and Rating Prediction.
>
> |Method|AmazonReviewTitle R-1 ↑|AmazonReviewTitle R-L ↑|AmazonReviewText R-1 ↑|AmazonReviewTitle R-L ↑|AmazonReviewRating MAE ↓|AmazonReviewRating RMSE ↓|
> |---|---|---|---|---|---|---|
> |Qwen3-8B-nothinking|0.172|0.163|0.242|0.173|0.242|0.549|
> |Qwen3-8B-thinking|0.179|0.165|0.182|0.135|0.282|0.599|
> |Qwen3-32B-nothinking|0.171|0.161|**0.247**|**0.174**|0.232|0.533|
> |Qwen3-32B-thinking|0.198|0.191|0.215|0.149|0.282|0.566|
> |GPT-4o|0.140|0.136|0.215|0.146|0.301|0.549|
> |Gemini-2.5-Pro|0.204|0.196|0.223|0.149|0.301|0.615|
> |TagPR (Our)|**0.210**|**0.202**|0.235|0.169|**0.222**|**0.511**|
>
> [2] Personalized Graph-Based Retrieval for Large Language Models, Arxiv, 2025.
>
> ---
> **Question 1**
>
> *Could you elaborate on how the choice of the 9 primary tags in the tagged reasoning chains dataset was determined?*
>
> **Response:**
>
> Thank you for pointing out the need for further clarification. We will revise the manuscript to more explicitly state that the 9 primary tags were determined through a bottom-up, data-driven process. The process involved three steps:
>
> - Open-ended Annotation: We utilized GPT-4o to generate diverse descriptive tags for reasoning chain segments. This exploratory step aimed to capture a comprehensive range of reasoning operations without predefined bias. The compressed prompt is as follows (Full prompts are provided in the Appendix D.1.2):
>
> > Role: You are an expert specializing in ... You will then break down this Chain-of-Thought into
> meaningful segments and assign an XML-style tag to each segment that best describes its
> function.
> > Task:
> > Based on the user-provided [Question] and the model-generated [Chain-of-Thought],
> > please complete the following steps:
> > 1.Analyze the Question and Chain-of-Thought: ...
> > 2.Segment the Chain-of-Thought: ...
> > 3.Tag Each Segment: ...
> > 4.Output the Result: ...
> > Tags should reflect function: ...
> > Maintain consistency: ...
> > Tags should be concise and clear: ...
> > Example: ...
> > Now, according to the rules above, please add tags to the [Question] and [Chain-of-Thought] provided below:
> > [Question]:
> > {question}
> > [Chain-of-Thought to be tagged]:
> > {chain of thought}
>
> - Semantic Clustering: We applied K-Means clustering to these initial tags to group semantically similar descriptions.
>
> - Tag Identification: From the resulting clusters, we identified 9 distinct tags that emerged as the most frequent and representative core cognitive steps. The full tags list is \<analyze_input\>, \<examine_examples\>, \<identify_patterns\>, \<evaluate_reference\>, \<compare_entities\>, \<synthesize_findings\>, \<make_decision\>, \<verify_conclusion\>, \<formulate_conclusion\>.

---

> ### Author Response · Authors · 2025-11-21
> **Response to Question 2 and Question 3**
>
> **Question 2**
>
> *Will the method benefit more from more careful choices on the reward composition?*
>
> **Response:**
>
> Thank you for this insightful suggestion regarding the sensitivity of reward composition.
>
> Our ablation study (Table 1 in Section 4.3) confirms that all three components are indispensable, as removing any single one leads to a performance drop. Our current weight assignment (α=0.8, β=0.8, γ=0.2) follows a specific design rationale: prioritizing structural integrity over personalization. High weights on reasoning and format ensure the model first establishes logically correct reasoning chains, while a lower weight on personalization prevents the model from sacrificing factuality to over-align with user preferences and reward hacking.
>
> Following your advice, we have included a hyperparameter sensitivity analysis. The results indicate that our current configuration yields the most balanced performance, validating our initial design intuition.
>
> |Method|LaMP-1 ACC ↑| LaMP-1 F1 ↑| LaMP-2 ACC ↑| LaMP-2 F1 ↑|LaMP-3 MAE ↓| LaMP-3 RMSE ↓| LaMP-4 R-1 ↑| LaMP-4 R-L ↑| LaMP-5 R-1 ↑| LaMP-5 R-L ↑| LaMP-7 R-1 ↑| LaMP-7 R-L ↑|
> |---|---|---|---|---|---|---|---|---|---|---|---|---|
> |$\alpha=1.0, \beta=1.0, \gamma=1.0$|0.797|0.797|0.580|0.552|0.226|0.271|0.226|0.208|0.540|0.469|0.563|0.504|
> |$\alpha=0.2, \beta=0.2, \gamma=0.8$|0.789|0.788|0.572|0.531|0.230|0.272|0.225|0.207|0.532|0.459|0.561|0.500|
> |$\alpha=0.8, \beta=0.2, \gamma=0.2$|0.789|0.789|0.575|0.536|0.229|0.272|0.227|0.208|0.539|0.466|0.564|0.505|
> |$\alpha=0.2, \beta=0.8, \gamma=0.2$|0.785|0.785|0.573|0.535|0.235|0.275|0.225|0.206|0.534|0.459|0.560|0.498|
> |$\alpha=0.8, \beta=0.8, \gamma=0.4$|0.801|0.800|0.594|0.553|**0.216**|**0.262**|0.231|0.212|**0.542**|0.470|0.564|0.505|
> |$\alpha=0.8, \beta=0.8, \gamma=0.2$ (Our)|**0.803**|**0.803**|**0.598**|**0.557**|0.218|0.263|**0.234**|**0.213**|**0.542**|**0.471**|**0.565**|**0.507**|
>
> ---
> **Question 3**
>
> *How well does the PRMU improve the model performance?*
>
> **Response:**
>
> Thank you for this insightful question. To quantify the performance improvements brought by the PRMU, we conducted a comprehensive ablation study comparing four settings: (1) Full PRMU, (2) PRMU w/o User Embeddings, (3) Untrained RM, and (4) w/o RM in Section 4.6.1.
>
>
> |Method|LaMP-1 ACC ↑| LaMP-1 F1 ↑| LaMP-2 ACC ↑| LaMP-2 F1 ↑|LaMP-3 MAE ↓| LaMP-3 RMSE ↓| LaMP-4 R-1 ↑| LaMP-4 R-L ↑| LaMP-5 R-1 ↑| LaMP-5 R-L ↑| LaMP-7 R-1 ↑| LaMP-7 R-L ↑|
> | --- | --- | --- | --- | --- | --- | --- | --- | --- | --- | --- | --- | --- |
> | w/o RM | 0.768 | 0.769 | 0.557 | 0.514 | 0.246 | 0.393 | 0.205 | 0.190 | 0.522 | 0.453 | 0.545 | 0.490 |
> | Untrained RM | 0.771 | 0.772 | 0.533 | 0.495 | 0.246 | 0.361 | 0.207 | 0.195 | 0.536 | 0.459 | 0.545 | 0.487 |
> | PRMU w/o UE | 0.784 | 0.784 | 0.581 | 0.541 | 0.231 | 0.299 | 0.215 | 0.197 | 0.536 | 0.467 | 0.558 | 0.501 |
> | PRMU | **0.803** | **0.803** | **0.598** | **0.557** | **0.218** | **0.263** | **0.234** | **0.213** | **0.542** | **0.471** | **0.565** | **0.507** |
>
> As shown, the results clearly demonstrate the effectiveness of our method: The full PRMU model consistently achieves the best performance, outperforming all other settings across all LaMP benchmark tasks.
>
> We would like to elaborate on the mechanism of improvement: The superiority of PRMU stems from its ability to provide fine-grained, user-specific reward signals.
> - From Generic to Personalized: Unlike untrained reward models that only judge generic quality, the PRMU is trained on personalized data to recognize "good personalized reasoning."
> - From Group to Individual: Crucially, the User Embeddings enable the reward signal to shift from understanding "general personalization" to capturing specific users' unique preferences, thereby guiding the model to generate content tailored to individuals.
>
> ---
>
> We truly appreciate your valuable feedback and the time you have dedicated to reviewing our work! We hope our responses have adequately addressed your concerns. We would be deeply grateful if you could consider raising your score, and we welcome any further discussion or questions you may have.

---

### Author Response · Authors · 2025-12-02
**Summary of Rebuttal**

Dear ACs and Reviewers:

We sincerely appreciate the constructive feedback provided by all reviewers and the time dedicated by the ACs under these special circumstances.

---
First, we are encouraged that the reviewers have provided positive assessments and recognized the core value of our work:

*   **State-of-the-Art Performance:** Reviewer `mHx1` and Reviewer `yWps` highlighted that TagPR achieves effective results, outperforming base models and larger proprietary LLMs on the LaMP and Dianping benchmark.
*   **Methodological Value:** Reviewer `mT6b` noted that our "Tagging the Thought" approach and the resulting dataset are valuable resources for the community. Reviewer `mHx1` found the SFT + multi-stage RL framework insightful.
*   **Interpretability:** Reviewer `yWps` acknowledged that structured, interpretable reasoning is a valuable direction compared to implicit reasoning.
*   **Well-written:** Reviewer `yWps` commended the paper for being well-written and clearly describing the complete reasoning chain construction and training process.

---
Based on these assessments, the reviewers raised insightful questions regarding generalization, reward design, and implementation details. We have conducted extensive additional experiments and analyses, which can be categorized into four main areas:

**Regarding Generalization and Datasets:**
To address the shared concern of Reviewer `mHx1` and Reviewer `yWps` regarding the model's effectiveness beyond the LaMP benchmark and Dianping benchmark:
*   We extended our evaluation to the **LongLaMP** benchmark. Crucially, we demonstrated that our distilled Tags and SFT-initialized model can be efficiently transferred to new tasks via RL without reconstructing massive reasoning chains, proving cost-efficiency and transferability.
*   We further evaluated TagPR on the **PGraphRAG** benchmark (Amazon Review subset), where it achieved superior performance despite extreme data sparsity.

**Regarding Methodology and Reward Design:**
*   In response to Reviewer `yWps`'s concern about Equation 5, we clarified the hierarchical gating mechanism of the reward design and provided ablation studies comparing hybrid vs. additive structures to validate the effectiveness of our reward design.
*   Addressing Reviewer `mT6b`'s query on the motivation for "tagging", we clarified how explicit tags enable process-level supervision and interpretability, preventing the model from learning spurious correlations.
*   We provided a detailed explanation of the "unshackling" logic in the Exploratory RL phase for Reviewer `mT6b`.

**Regarding Hyperparameters and Sensitivity:**
*   Following suggestions from Reviewer `mHx1` and Reviewer `mT6b`, we conducted sensitivity analyses on reward weights ($\alpha, \beta, \gamma$), demonstrating the performance under different reward weights.
*   Specifically responding to Reviewer `yWps`'s request, we performed a Control Variable Analysis, isolating each reward component. This analysis confirmed the distinct roles of Verifiable, Tag, and Personalization rewards, validating our design rationale.

**Regarding Interpretability and Standards:**
*   To address Reviewer `yWps`'s questions on reasoning rationality, we provided additional qualitative case studies and analyzed the logical hierarchy of generated tags.
*   We standardized mathematical notations, explicitly clarified evaluation protocols (LaMP official metrics), and supply Implementation Guide to address Reviewer `mT6b` and Reviewer `yWps`'s comments on reproducibility.

**In summary**, we have provided comprehensive responses to all reviewers’ concerns, further refining the presentation and reinforcing the significance of our contributions. In the revised manuscript, we have incorporated additional experimental results and key discussions, with all new content highlighted in blue.

---
We are pleased to report that we have actively engaged with Reviewer `yWps`, who explicitly acknowledged our Control Variable Analysis, **confirmed that their concerns were resolved, and increased their rating**. Regrettably, due to the early closure of the rebuttal period, we did not receive feedback from Reviewers `mHx1` and `mT6b`. However, we remain confident that our comprehensive responses have effectively addressed their core concerns. This is particularly true for Reviewer `mT6b`, whose concerns stemmed largely from a lack of familiarity with the personalization field, which we have now extensively clarified.

---
Once again, we express our deepest gratitude to the reviewers and ACs for their invaluable insights and dedication in improving our work!

Best regards,

The Authors

---

### Note · Authors · 2026-01-05

I have read and agree with the venue's withdrawal policy on behalf of myself and my co-authors.